# Beyond Independent Genes: Learning Module-Inductive Representations for Single-Cell Gene Perturbation Prediction

**Jiafa Ruan** [1 2] **Ruijie Quan** [1 2] **Liyang Xu** [1 2] **Zongxin Yang** [3] **Yi Yang** [1 2]

## Abstract

Predicting transcriptional responses to genetic perturbations is a central problem in functional genomics. In practice, perturbation responses are rarely gene-independent but instead manifest as coordinated, program-level transcriptional changes among functionally related genes. However, most existing methods do not explicitly model such coordination, due to gene-wise modeling paradigms and reliance on static biological priors that cannot capture dynamic program reorganization. To address these limitations, we propose **scBIG**, a module-inductive perturbation prediction framework that explicitly models coordinated gene programs. scBIG induces coherent gene programs from data via Gene-Relation Clustering, captures inter-program interactions through a Gene-Cluster-Aware Encoder, and preserves modular coordination using structure-aware alignment objectives. These structured representations are then modeled using conditional flow matching to enable flexible and generalizable perturbation prediction. Extensive experiments on multiple single-cell perturbation benchmarks show that scBIG consistently outperforms state-of-the-art methods, particularly on unseen and combinatorial perturbation settings, achieving an average improvement of 6.7% over the strongest baselines. The code is available at https://github.com/ttruan2426-dot/scBIG.

[1]The State Key Lab of Brain-Machine Intelligence, Zhejiang University, Hangzhou, Zhejiang, China [2]ReLER Lab, College of Artificial Intelligence, Zhejiang University, Hangzhou, Zhejiang, China [3]Department of Biomedical Informatics, Harvard Medical School, Boston, MA, USA. Correspondence to: Zongxin Yang <Zongxin_Yang@hms.harvard.edu>.

*Proceedings of the 43rd International Conference on Machine Learning*, Seoul, South Korea. PMLR 306, 2026. Copyright 2026 by the author(s).

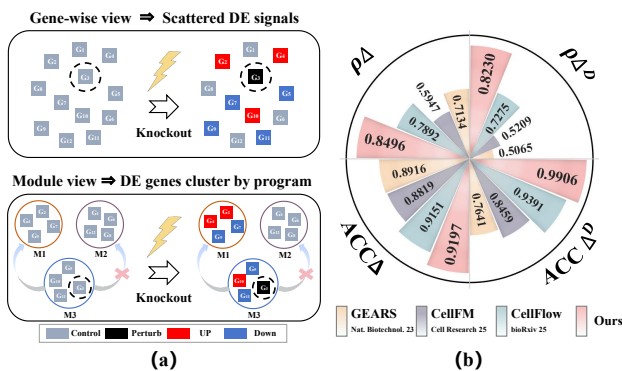

*Figure 1.* **(a)**: Comparison between gene-wise view and module view in perturbation responses. **Black** indicates the perturbed gene, red indicates upregulation, and blue indicates downregulation. **(b)**: Quantitative comparison of our method and state-of-the-art approaches on the Norman additive split.

## 1. Introduction

Predicting transcriptional responses to gene perturbations is a core problem in functional genomics, with broad applications in gene function discovery (Dixit et al., 2016; Replogle et al., 2022), regulatory mechanism analysis (Aibar et al., 2017; Theodoris et al., 2023), and therapeutic development (Lotfollahi et al., 2023; Qi et al., 2024). Recent advances in CRISPR-based perturbation technologies and single-cell RNA sequencing enable high-resolution measurements of cellular responses to gene perturbations (Dixit et al., 2016; Jaitin et al., 2016; Datlinger et al., 2017). However, the combinatorial explosion of possible perturbation conditions makes exhaustive experiments infeasible, motivating the development of accurate and generalizable *in silico* perturbation prediction models.

A fundamental biological characteristic of perturbation responses is that they are rarely gene-independent (Dixit et al., 2016). In practice, genetic perturbations induce structured, program-level transcriptional responses (Subramanian et al., 2005), in which groups of functionally related genes (often corresponding to biological pathways such as cell-cycle regulation or stress response) are co-regulated to execute specific cellular processes. Particularly, these modular responses are pronounced in differentially expressed genes and under combinatorial perturbations (Norman et al., 2019),

where interactions between gene programs, rather than isolated genes, govern cellular outcomes. Accurately modeling these coordinated, module-level behaviors is therefore essential for robust perturbation prediction.

Although recent approaches have made substantial progress (Lotfollahi et al., 2019; Wu et al., 2022), they do not explicitly model coordinated program-level responses in gene perturbation prediction. This limitation arises from two complementary factors. **i) Gene-wise modeling neglects program-level coordination.** Many foundation and dynamic generative models (Cui et al., 2024; Hao et al., 2024; Yang et al., 2024; Zeng et al., 2025; Chi et al., 2025a; Klein et al., 2025) represent gene expression in a flat, gene-wise manner or compress it into unstructured latent spaces. While effective for modeling marginal gene expression, such representations lack explicit mechanisms to capture coordination among groups of functionally related genes, making it difficult to model program-level transcriptional responses. **ii) Static biological priors cannot capture dynamic program coordination.** Graph-based approaches (Kamimoto et al., 2023; Bereket & Karaletsos, 2023; Roohani et al., 2024; Chi et al., 2025b) attempt to incorporate gene relations through predefined priors such as Gene Ontology (Consortium, 2004), protein–protein interaction networks (Szklarczyk et al., 2019), or gene regulatory graphs (Karlebach & Shamir, 2008). However, these priors are typically static, incomplete, and context-agnostic, limiting their ability to model the dynamic reorganization and coordination of gene programs across different cell types and perturbation conditions (Milano et al., 2022).

To address these challenges, a perturbation model should both induce gene programs beyond flat gene-wise representations and model their coordinated interactions in a context-adaptive manner, without relying on fixed biological graphs. Building on this insight, we propose **scBIG**, a module-inductive perturbation prediction framework designed to explicitly model coordinated gene programs. scBIG addresses the aforementioned challenges through a three-stage design of ⟨module induction, structured generative modeling, structure-aware alignment⟩. **i) Gene-Relation Clustering (GRC) (§2.1).** To move beyond flat gene-wise representations, scBIG first induces biologically coherent gene programs directly from data. GRC adaptively partitions genes into functional modules by integrating semantic embeddings from pretrained foundation models with high-confidence protein–protein interaction priors, providing a program-level inductive scaffold. **ii) Gene-Cluster-Aware Encoder (GCAE) and Conditional Flow Matching (§2.2, §2.3).** Building on the induced programs, we introduce a hierarchical GCAE that explicitly models high-order interactions among gene programs, enabling structured reasoning beyond independent genes. These structured representations are then modeled within a *conditional flow matching*

framework, which serves as a flexible generative backbone for perturbation response prediction. **iii) Structure-Aware Alignment (§2.4).** To ensure that induced program coordination is preserved during generation, we further introduce a structure-aware alignment mechanism comprising *Cluster Correlation Alignment* and *Pathway-informed Optimal Transport*. These objectives enforce consistency at both the module and pathway levels, encouraging biologically coherent and coordinated transcriptional responses.

Extensive experiments on multiple single-cell perturbation benchmarks demonstrate that scBIG consistently outperforms state-of-the-art methods, particularly on unseen and combinatorial perturbation settings, *e.g.*, it surpasses 13 leading methods, delivering an average performance gain of 6.7% over the best-performing baselines on key metrics.

Our main contributions are summarized as follows:

- To the best of our knowledge, this work is among the first to explicitly formalize a *module-level* inductive bias for *generative* perturbation prediction, moving beyond unstructured gene-wise modeling toward coordinated functional programs.

- We present **scBIG**, a module-inductive framework that induces gene programs from data (via GRC), performs hierarchical reasoning over inter-program interactions (via a GCAE), and enforces structure-preserving objectives to maintain biological coherence during generation (via Structure-Aware Alignment).

- Extensive evaluations on multiple single-cell perturbation benchmarks show that scBIG consistently outperforms 13 strong baselines, with particularly large gains on out-of-distribution generalization and combinatorial perturbation settings.

## 2. Method

**Overview.** As shown in Fig. 2, scBIG is a module-inductive generative framework for predicting transcriptional responses to genetic perturbations with high structural fidelity. It is designed to explicitly capture program structure and preserve coordinated responses during generation. The framework consists of three components. These components work in a complementary manner, from inducing gene modules to generating and aligning structure-consistent perturbation responses. (i) Gene-Relation Clustering (§2.1) adaptively partitions the gene space into biologically coherent modules by fusing foundation-model semantics with high-confidence PPI priors. (ii) Generative Modeling via GCAE(§2.2) uses a Gene-Cluster-Aware Encoder to capture inter-module dependencies and produce structured latent representations for a conditional flow-matching backbone (§2.3). (iii) Structure-Aware Alignment (§2.4) enforces biological consistency via *Cluster Correlation Alignment* and

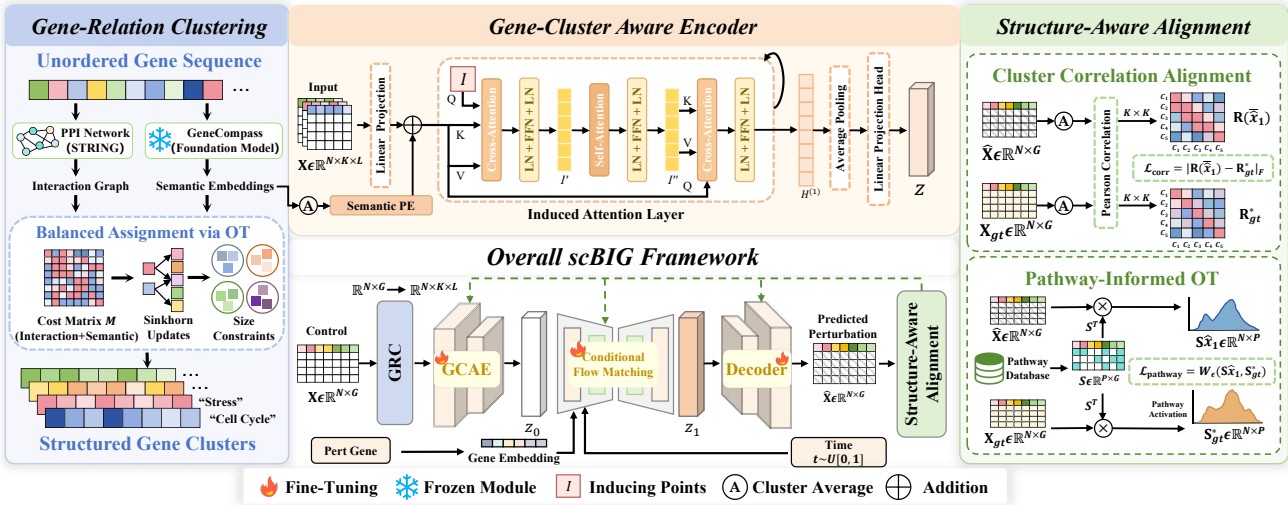

*Figure 2.* **Overview of the scBIG framework.** (§2) **(Left) Gene-Relation Clustering (GRC)** partitions the unordered gene space into $K$ biologically coherent modules via optimal transport, integrating semantic embeddings from foundation models with high-confidence PPI priors. **(Middle) Generative Backbone.** The framework encodes cells using the **Gene-Cluster-Aware Encoder (GCAE)**, which captures high-order inter-module interactions through a bottleneck attention mechanism with inducing points **(Top)**. These latent representations guide a **Conditional Flow Matching** module **(Bottom)** to model the continuous transition from control ($z_0$) to perturbed ($z_1$) states. **(Right) Structure-Aware Alignment.** To ensure phenotypic fidelity, the entire pipeline is jointly optimized with two structural regularizers: **Cluster Correlation Alignment**, which preserves module-level co-expression patterns, and **Pathway-informed Optimal Transport**, which aligns predicted responses with canonical biological pathways.

*Pathway-informed Optimal Transport.*

Together, these components yield predictions that are accurate at the expression level while preserving coordinated, program-level structure. Implementation details, including default hyperparameter settings and full algorithmic procedures, are provided in §A.4.1.

### 2.1. Gene-Relation Clustering (GRC)

The unordered and high-dimensional nature of single-cell gene expression data presents a significant challenge for revealing meaningful regulatory structures. To introduce a structured inductive bias, we first partition the $G$ genes into $K$ functionally coherent clusters, $\mathcal{C} = \{c_1, \ldots, c_K\}$, capturing both semantic similarity and biological interactions.

**Integrating Semantic and Biological Priors.** Our Gene-Relation Clustering (GRC) approach constructs this gene partition by solving a balanced assignment problem using Optimal Transport (OT). This framework integrates two complementary priors: First, the semantic representation of genes is derived from a pre-trained foundation model (*e.g.*, GeneCompass (Yang et al., 2024)). Specifically, we obtain a gene embedding matrix $\mathbf{E} = [E_1, \ldots, E_G]^\top \in \mathbb{R}^{G \times d}$, where each gene embedding $E_i$ encodes co-expression patterns and context-specific regulatory information.Second, a binary Protein-Protein Interaction (PPI) adjacency matrix $\mathbf{A}^{\text{PPI}} \in \{0,1\}^{G \times G}$ is constructed by thresholding interaction scores from the STRING (Szklarczyk et al., 2019) database at 700 to retain high-confidence interactions.

**Cost Matrix Formulation.** To jointly leverage these two priors, we define cluster centroids $\{\mu_k\}_{k=1}^K$ in the semantic space, and construct a combined cost matrix $\mathbf{C} \in \mathbb{R}^{G \times K}$, where each entry is computed as:

$$C_{ik} = D^{\text{sem}}(E_i, \mu_k) + D^{\text{PPI}}(i, k), \tag{1}$$

where $D^{\text{sem}}(E_i, \mu_k) = 1 - \cosine(E_i, \mu_k)$ measures the semantic dissimilarity between gene $i$ and cluster $k$, and $D^{\text{PPI}}(i, k) = 1 - \frac{1}{|c_k|} \sum_{j \in c_k} A_{ij}^{\text{PPI}}$ penalizes low PPI coherence within cluster $k$.

**Balanced Assignment via Optimal Transport.** We solve the assignment problem by minimizing the transport cost $\langle \mathbf{\Pi}, \mathbf{C} \rangle$ subject to uniform marginal constraints $a = \frac{1}{G} \mathbf{1}_G$ and $b = \frac{1}{K} \mathbf{1}_K$. This OT formulation imposes an approximately balanced soft assignment. To handle the dependency of $D^{\text{PPI}}$ on cluster membership, GRC is solved *offline* through an alternating optimization: we update the transport plan $\mathbf{\Pi}$ via the Sinkhorn algorithm, followed by a capacity-constrained rounding step to ensure a strictly balanced hard partition (i.e., $|c_k| = G/K$). The balanced constraint is an architectural regularizer for stable chunk-wise attention, rather than a claim that biological programs are naturally equal-sized; it prevents oversized clusters from diluting attention capacity.

**Module-Structured Input Construction.** Each gene is assigned to cluster $k_i = \arg\max_k \mathbf{\Pi}_{ik}^*$. The resulting clusters are fixed to reorder the raw data $\mathbf{X} \in \mathbb{R}^{N \times G}$ into a structured representation $\mathbf{X} \in \mathbb{R}^{N \times K \times L}$, where $L = G/K$.

This transformation provides the hierarchical input necessary for downstream inter-module modeling.

## 2.2. Gene-Cluster Aware Encoder (GCAE)

To bridge high-resolution transcriptomics with structured modules, GCAE operates on $K$ gene clusters defined by GRC. For each cluster $k$, we derive a comprehensive module representation $\mathbf{h}_k^{(0)} \in \mathbb{R}^D$ through a dual-stream fusion:

$$\mathbf{h}_k^{(0)} = \text{Proj}_{\text{exp}}(\mathbf{x}_{c_k}) + \text{Proj}_{\text{sem}}\left(\frac{1}{|c_k|} \sum_{g \in c_k} E_g\right), \quad (2)$$

where $\mathbf{x}_{c_k} \in \mathbb{R}^L$ (with $L = G/K$) is the raw expression vector of genes in cluster $k$, and $E_g$ are pre-trained foundation model embeddings. This design projects both local expression patterns and global semantic priors into a shared $D$-dimensional latent space. The resulting $\mathbf{H}^{(0)} \in \mathbb{R}^{K \times D}$ is augmented with relational position encodings as the input for inter-module modeling.

**Inter-Module Modeling via Latent Bottleneck.** To capture regulatory dependencies, GCAE employs a Perceiver-style bottleneck (Jaegle et al., 2021) with $M$ learnable *Inducing Points* $\mathbf{I} \in \mathbb{R}^{M \times D}$ ($M < K$). The inducing points are model-level parameters initialized once, shared across samples, and jointly optimized during training; they act as latent bottleneck slots rather than fixed biological entities, so $M$ controls bottleneck capacity rather than a biological quantity. The encoding follows a "Compress-Process-Broadcast" paradigm:

$$\begin{aligned}
\mathbf{I}' &= \text{LN}(\mathbf{I} + \text{CrossAttn}(\mathbf{I}, \mathbf{H}^{(0)})), \\
\mathbf{I}'' &= \text{LN}(\mathbf{I}' + \text{SelfAttn}(\mathbf{I}')), \quad (3) \\
\mathbf{H}^{(1)} &= \text{LN}(\mathbf{H}^{(0)} + \text{CrossAttn}(\mathbf{H}^{(0)}, \mathbf{I}'')),
\end{aligned}$$

where $\mathbf{I}'$ aggregates information from all modules, $\mathbf{I}''$ captures global latent interactions, and $\mathbf{H}^{(1)}$ represents the updated cellular state.

$$\mathbf{Z} = \text{Linear}\left(\frac{1}{K} \sum_{k=1}^{K} \mathbf{H}_k^{(1)}\right), \quad (4)$$

where $\mathbf{Z} \in \mathbb{R}^D$ is the final latent embedding utilized for perturbation flow matching and reconstruction. This design ensures that the cell-level representation encapsulates the collaborative dynamics of all functional gene modules.

**Latent Manifold Pre-training.** The encoder-decoder pair is pre-trained via reconstruction to ensure a robust latent manifold. Specifically, we use a reconstruction loss defined as:

$$\mathcal{L}_{\text{recon}} = \frac{1}{N} \sum_{i=1}^{N} \|\hat{\mathbf{x}}_i - \mathbf{x}_i\|_2^2, \quad (5)$$

where $\mathbf{x}_i$ represents the $i$-th input expression vector from the batch, and $\hat{\mathbf{x}}_i$ is the corresponding reconstructed expression vector. This loss encourages the model to learn a meaningful and robust latent representation of the input gene expression data.

## 2.3. Conditional Flow Matching Backbone

**Perturbation Vector Field Regression.** We formalize perturbation prediction as learning a conditional probability path between the latent states of control and perturbed cells. Let $\mathbf{z}_0 = \text{GCAE}(\mathbf{x}_{\text{ctrl}})$ and $\mathbf{z}_1 = \text{GCAE}(\mathbf{x}_{\text{pert}})$ denote the latent embeddings of unperturbed and perturbed cells, respectively, as derived from the pre-trained encoder. We define a time-varying vector field $v_\theta(\mathbf{z}_t, t, \mathbf{c})$ parameterized by a neural network, where $\mathbf{c}$ encodes the perturbation via ESM2 (Lin et al., 2023) embeddings (Wang et al., 2024). The model is trained to regress the vector field:

$$\mathcal{L}_{\text{flow}} = \mathbb{E}_{t, \mathbf{z}_0, \mathbf{z}_1} \|v_\theta(\mathbf{z}_t, t, \mathbf{c}) - (\mathbf{z}_1 - \mathbf{z}_0)\|^2, \quad (6)$$

where $\mathbf{z}_t = (1-t)\mathbf{z}_0 + t\mathbf{z}_1$ is the interpolation path.

To ensure the decoded expression $\hat{\mathbf{x}}_1 = \text{Dec}(\hat{\mathbf{z}}_1)$ adheres to biological priors, we introduce two structured consistency objectives through the differentiable encode-flow-decode pipeline.

## 2.4. Structure-Aware Alignment Objectives

**Cluster Correlation Alignment.** To preserve the functional co-expression patterns between gene modules, we constrain the inter-cluster correlation structure of the predicted expression profile. Specifically, let $\mathbf{R}(\hat{\bar{\mathbf{x}}}1) \in \mathbb{R}^{K \times K}$ be the Pearson correlation matrix of the cluster-aggregated expressions. We then align it with the ground truth via the loss:

$$\mathcal{L}_{\text{corr}} = \|\mathbf{R}(\hat{\bar{\mathbf{x}}}_1) - \mathbf{R}_{gt}^*\|_F, \quad (7)$$

where $\mathbf{R}_{gt}^*$ is the precomputed target correlation for perturbation $p$. This constrains the model to capture higher-order regulatory dependencies between gene modules. **Pathway-informed Optimal Transport.** To enforce phenotypic validity, we match the distribution of predicted pathway activations to the ground truth. Specifically, we leverage a pathway membership matrix $\mathbf{S} \in \{0,1\}^{P \times G}$ mapping $G$ genes to $P$ pre-defined biological pathways from Reactome database (Fabregat et al., 2018). We then compute the Sinkhorn distance in the pathway-quotient space:

$$\mathcal{L}_{\text{pathway}} = \mathcal{W}_\epsilon\left(\mathbf{S}\hat{\mathbf{x}}_1, \mathbf{S}_{gt}^*\right), \quad (8)$$

where $\mathbf{S}_{gt}^*$ denotes target activations. This ensures the generated cells lie within biologically plausible manifolds.

The total objective is:

$$\mathcal{L} = \mathcal{L}_{\text{flow}} + \lambda_{\text{corr}}\mathcal{L}_{\text{corr}} + \lambda_{\text{pathway}}\mathcal{L}_{\text{pathway}}, \quad (9)$$

optimized end-to-end with target statistics cached for efficiency.

*Table 1.* Comparison of different methods across evaluation metrics on the Norman additive split (§3.2); Type abbreviations: Sta.=Statistical, Found.=Foundation, Gra.=Graph-based, Gen.=Generative.

| Type | Method | Public | $\rho\Delta\uparrow$ | $\rho\Delta^D\uparrow$ | ACC$\Delta\uparrow$ | ACC$\Delta^D\uparrow$ | DES↑ | PDS↑ | L2↓ | MSE↓ | MAE↓ |
|---|---|---|---|---|---|---|---|---|---|---|---|
| Sta. | Control | – | – | – | – | – | – | 0.5000 | 9.25 | 0.0487 | 0.1720 |
| | Linear | – | 0.6211 | 0.7075 | 0.8007 | 0.8906 | 0.3192 | 0.5363 | 6.76 | 0.0265 | 0.1233 |
| | Linear-scGPT (Ahlmann-Eltze et al., 2025) | *Nature Methods'25* | 0.6925 | 0.7514 | 0.8424 | 0.9688 | 0.5023 | 0.7913 | 5.31 | 0.0155 | 0.0942 |
| Found. | scGPT (Cui et al., 2024) | *Nature Methods'24* | 0.4408 | 0.4416 | 0.8125 | 0.4839 | 0.2504 | 0.5022 | 7.35 | 0.0296 | 0.1285 |
| | scFoundation (Hao et al., 2024) | *Nature Methods'24* | 0.6813 | 0.4778 | 0.8768 | 0.7453 | 0.5409 | 0.7994 | 4.91 | 0.0138 | 0.0852 |
| | GeneCompass (Yang et al., 2024) | *Cell Research'24* | 0.6897 | 0.4810 | 0.8916 | 0.7484 | 0.5948 | 0.8024 | 4.77 | 0.0124 | 0.0808 |
| | CellFM (Zeng et al., 2025) | *Cell Research'25* | 0.5947 | 0.5209 | 0.8819 | 0.8459 | 0.7550 | 0.7419 | 5.41 | 0.0161 | 0.0889 |
| Gra. | GEARS (Roohani et al., 2024) | *Nat. Biotechnol.'23* | 0.7134 | 0.5065 | 0.8916 | 0.7641 | 0.6220 | 0.8155 | 4.59 | 0.0117 | 0.0788 |
| | CellOracle (Kamimoto et al., 2023) | *Nature'23* | 0.0437 | 0.2567 | 0.4942 | 0.4536 | 0.0319 | 0.5277 | 11.1 | 0.0691 | 0.2001 |
| | GraphVCI (Bereket & Karaletsos, 2023) | *ICLR'23* | 0.5468 | 0.5722 | 0.8034 | 0.7905 | 0.6495 | 0.5151 | 6.68 | 0.0258 | 0.1177 |
| Gen. | samsVAE (Wu et al., 2022) | *NeurIPS'23* | 0.6497 | 0.7750 | 0.8470 | 0.9658 | 0.6138 | 0.6689 | 6.87 | 0.0329 | 0.1067 |
| | STATE (Adduri et al., 2025) | *bioRxiv'25* | 0.2439 | 0.2628 | 0.8023 | 0.4141 | 0.4891 | 0.5171 | 9.02 | 0.0421 | 0.1343 |
| | CellFlow (Klein et al., 2025) | *bioRxiv'25* | 0.7892 | 0.7275 | 0.9151 | 0.9391 | 0.8113 | 0.7581 | 4.54 | 0.0114 | 0.0775 |
| | scBIG (Ours) | *Proposed* | 0.8496 | 0.8230 | 0.9197 | 0.9906 | 0.8593 | 0.8548 | 3.92 | 0.0091 | 0.0689 |

# 3. Experiments

## 3.1. Experimental Setup

**Datasets and Scenarios.** We evaluate **scBIG** on two genetic perturbation benchmarks, Norman (Norman et al., 2019) and Replogle2022_rpe1 (Replogle et al., 2022), and further test drug perturbations on ComboSciPlex. For genetic perturbations, we consider two settings. **(i) Additive setting:** we evaluate on *unseen dual-gene* perturbations formed by pairing two genes that are each observed in training, testing combinatorial generalization via composition. **(ii) Holdout setting:** we hold out a subset of genes entirely from training and evaluate zero-shot on perturbations involving these unseen genes, including *unseen single-gene* and *unseen dual-gene* (two unseen genes) cases. Detailed metric definitions, data processing, and split construction are provided in §A.4.

**Comparison Baselines.** We systematically compare **scBIG** against 13 state-of-the-art baselines across four paradigms: (1) **Foundation Models**, including **scGPT** (Cui et al., 2024), **scFoundation** (Hao et al., 2024), **GeneCompass** (Yang et al., 2024), and **CellFM** (Zeng et al., 2025); (2) **Graph-based Models**, such as **GEARS** (Roohani et al., 2024), **CellOracle** (Kamimoto et al., 2023), and **GraphVCI** (Bereket & Karaletsos, 2023); (3) **Generative Models** that operate in latent spaces, including **samsVAE** (Wu et al., 2022), **STATE** (Adduri et al., 2025), and **CellFlow** (Klein et al., 2025); and (4) **Statistical Methods**, comprising **Control**, **Linear**, and **Linear-scGPT** (Ahlmann-Eltze et al., 2025).

## 3.2. Benchmarking Performance on the Norman Dataset

**Additive Split.** As summarized in Table 1, **scBIG** establishes a new state-of-the-art across all evaluated metrics. Beyond achieving the lowest absolute errors in terms of L2, MSE, and MAE, our framework demonstrates a superior ability to capture perturbation-induced distributional shifts. Specifically, **scBIG** surpasses the advanced baseline *CellFlow* by **7.7%** on $\rho\Delta$ and *GeneCompass* by **6.5%** on PDS. Furthermore, the **5.9%** improvement over *CellFlow* on the DES metric underscores our model's precision in identifying differentially expressed genes (DEGs), validating the efficacy of our module-level inductive bias for predicting unseen combinatorial effects. As shown in Table 13, distribution-level metrics further confirm stronger perturbation-conditional recovery on this split: scBIG improves Discr. Cos while reducing both E-Dist and Wasserstein discrepancy. This indicates that the generated cells better match both perturbation direction and the full conditional response distribution. Across perturbations, the gains over CellFlow remain statistically significant under paired Wilcoxon tests, with mean ± SE and p-values reported in §A.3.2.

**Holdout Split.** Table 2 highlights the substantial advantages of **scBIG** in extrapolating to entirely unseen perturbations. On the unseen single-perturbation task, our method achieves significant relative gains over *CellFlow*, notably **38.0%** on $\rho\Delta^D$ and **12.4%** on PDS. In the even more challenging unseen double-perturbation task, while *Linear-scGPT* achieves competitive MSE, it does so at the expense of biological fidelity and DEG-specific accuracy. In contrast, **scBIG** outperforms *Linear-scGPT* by a wide margin—**20.6%** on $\rho\Delta^D$ and **35.9%** on DES. These results collectively affirm that our module-structured representations better preserve the functional gene programs that are often lost in global error-minimization approaches.

## 3.3. Validation on the RPE1 Dataset

As demonstrated in Table 3, scBIG establishes a new state-of-the-art on the RPE1 dataset, confirming that its advantages generalize to diverse cellular contexts beyond the K562 cell line. Our framework significantly outperforms all competing methods across nearly every metric. Notably, scBIG surpasses the previous leading method, CellFlow, by 25.7% on $\rho\Delta$ and 34.9% on $\rho\Delta^D$, showcasing its su-

*Table 2.* Comparison of different methods across evaluation metrics on the Norman holdout split (§3.2); Type abbreviations: Sta.=Statistical, Found.=Foundation, Gra.=Graph-based, Gen.=Generative.

| Type | Method | Public | $\rho\Delta\uparrow$ | $\rho\Delta^D\uparrow$ | ACC$\Delta\uparrow$ | ACC$\Delta^D\uparrow$ | DES↑ | PDS↑ | L2↓ | MSE↓ | MAE↓ |
|---|---|---|---|---|---|---|---|---|---|---|---|
| | | | *Single Gene Perturbation* | | | | | | | | |
| Sta. | Control | – | – | – | – | – | – | 0.5000 | 4.33 | 0.0108 | 0.0685 |
| | Linear | – | 0.5924 | 0.6021 | 0.7695 | 0.8476 | 0.4548 | 0.5119 | 3.46 | 0.0067 | 0.0516 |
| | Linear-scGPT (Ahlmann-Eltze et al., 2025) | *Nature Methods'25* | 0.6072 | 0.6352 | 0.7620 | 0.8095 | 0.4584 | 0.6571 | 3.60 | 0.0072 | 0.0575 |
| Found. | scGPT (Cui et al., 2024) | *Nature Methods'24* | 0.5172 | 0.5358 | 0.7509 | 0.8286 | 0.5987 | 0.5048 | 3.58 | 0.0071 | 0.0530 |
| | scFoundation (Hao et al., 2024) | *Nature Methods'24* | 0.4453 | 0.3144 | 0.6882 | 0.7429 | 0.3897 | 0.6619 | 3.65 | 0.0079 | 0.0568 |
| | GeneCompass (Yang et al., 2024) | *Cell Research'24* | 0.5307 | 0.3423 | 0.7006 | 0.7429 | 0.4543 | 0.5857 | 3.44 | 0.0066 | 0.0546 |
| | CellFM (Zeng et al., 2025) | *Cell Research'25* | 0.4331 | 0.5226 | 0.6708 | 0.7119 | 0.5318 | 0.5524 | 3.83 | 0.0086 | 0.0600 |
| Gra. | GEARS (Roohani et al., 2024) | *Nat. Biotechnol.'23* | 0.4539 | 0.5345 | 0.7059 | 0.8024 | 0.3968 | 0.4905 | 3.77 | 0.0078 | 0.0574 |
| | GraphVCI (Bereket & Karaletsos, 2023) | *ICLR'23* | 0.3902 | 0.1865 | 0.7054 | 0.5583 | 0.1395 | 0.5000 | 3.28 | 0.0066 | 0.0500 |
| Gen. | samsVAE (Wu et al., 2022) | *NeurIPS'23* | 0.4778 | 0.4982 | 0.7050 | 0.8310 | 0.3366 | 0.3452 | 3.88 | 0.0081 | 0.0614 |
| | STATE (Adduri et al., 2025) | *bioRxiv'25* | 0.3085 | 0.5207 | 0.7600 | 0.5763 | 0.5712 | 0.5048 | 5.94 | 0.0177 | 0.0737 |
| | CellFlow (Klein et al., 2025) | *bioRxiv'25* | 0.5930 | 0.4736 | 0.7778 | 0.8119 | 0.6280 | 0.6643 | 3.40 | 0.0076 | 0.0535 |
| | scBIG (Ours) | *Proposed* | **0.6507** | **0.6537** | **0.7876** | **0.8548** | **0.6333** | **0.7467** | **3.26** | **0.0065** | **0.0491** |
| | | | *Double Genes Perturbation* | | | | | | | | |
| Sta. | Control | – | – | – | – | – | – | 0.5000 | 6.02 | 0.0200 | 0.0984 |
| | Linear | – | 0.7179 | 0.8277 | 0.8240 | 0.9200 | 0.5637 | 0.5000 | 4.31 | 0.0105 | 0.0656 |
| | Linear-scGPT (Ahlmann-Eltze et al., 2025) | *Nature Methods'25* | 0.7351 | 0.7230 | 0.8282 | 0.9100 | 0.6118 | 0.7000 | 3.91 | **0.0080** | 0.0626 |
| Found. | scGPT (Cui et al., 2024) | *Nature Methods'24* | 0.6350 | 0.4719 | 0.8068 | 0.8333 | 0.4456 | 0.5000 | 4.70 | 0.0124 | 0.0726 |
| | scFoundation (Hao et al., 2024) | *Nature Methods'24* | 0.6605 | 0.4445 | 0.7964 | 0.9200 | 0.6538 | 0.6500 | 3.91 | 0.0083 | 0.0589 |
| | GeneCompass (Yang et al., 2024) | *Cell Research'24* | 0.6437 | 0.6866 | 0.7471 | 0.7400 | 0.5905 | 0.6000 | 4.37 | 0.0108 | 0.0723 |
| | CellFM (Zeng et al., 2025) | *Cell Research'25* | 0.5615 | 0.8151 | 0.7072 | 0.7600 | 0.6746 | 0.5000 | 4.89 | 0.0132 | 0.0804 |
| Gra. | GEARS (Roohani et al., 2024) | *Nat. Biotechnol.'23* | 0.5087 | 0.6578 | 0.7130 | 0.8300 | 0.4156 | 0.5000 | 5.00 | 0.0138 | 0.0788 |
| | GraphVCI (Bereket & Karaletsos, 2023) | *ICLR'23* | 0.4290 | 0.5429 | 0.6889 | 0.6533 | 0.4003 | 0.5048 | 4.61 | 0.0114 | 0.0725 |
| Gen. | samsVAE (Wu et al., 2022) | *NeurIPS'23* | 0.6267 | 0.6402 | 0.7539 | 0.8800 | 0.5183 | 0.6500 | 4.66 | 0.0118 | 0.0742 |
| | STATE (Adduri et al., 2025) | *bioRxiv'25* | 0.3786 | 0.6379 | 0.8126 | 0.5800 | 0.7097 | 0.5000 | 5.94 | 0.0177 | 0.0737 |
| | CellFlow (Klein et al., 2025) | *bioRxiv'25* | 0.7404 | 0.6774 | 0.8388 | 0.8400 | 0.7636 | 0.4500 | 4.12 | 0.0097 | 0.0621 |
| | scBIG (Ours) | *Proposed* | **0.8026** | **0.8719** | **0.8552** | **0.9700** | **0.8313** | **0.7440** | **3.68** | 0.0083 | **0.0554** |

perior ability to capture both global expression changes and key DEG-specific effects. This enhanced precision is further evidenced by a 1.5% improvement in PDS and top scores in ACC$\Delta$ and ACC$\Delta^D$. Moreover, our model consistently minimizes prediction error, reducing L2, MSE, and MAE by 5.6%, 12.6%, and 4.9% respectively, relative to CellFlow. Collectively, these results validate that the biological structure incorporated into scBIG provides a robust and applicable inductive bias for gene perturbation modeling.

## 3.4. Validation on Drug Molecule Perturbation

As shown in Table 4, scBIG consistently achieves the best performance across all evaluation metrics on the ComboSciPlex drug molecule perturbation benchmark, demonstrating its strong generalization capability beyond genetic perturbations. For drug perturbations, we use pretrained ChemBERTa embeddings (Chithrananda et al., 2020) as the molecular condition injected into scBIG. Compared with the strongest competing baseline for each metric, scBIG improves $r$ by 3.4% over CPA and $r_{DEG}$ by 0.4% over scGPT, indicating more accurate recovery of both global transcriptional responses and DEG-specific drug effects. The advantage is more pronounced in perturbation direction consistency, where scBIG improves Discr. Cos by 2.7% over CPA. Moreover, scBIG also achieves the lowest prediction error and distributional discrepancy, reducing MSE by 2.2% and E-Dist by 1.9% relative to CellFlow, while fur-

ther decreasing Wasserstein by 4.0% compared with scGPT. These results suggest that the biological inductive biases incorporated into scBIG are not limited to gene perturbation modeling, but also provide a robust and transferable framework for predicting drug-induced cellular responses.

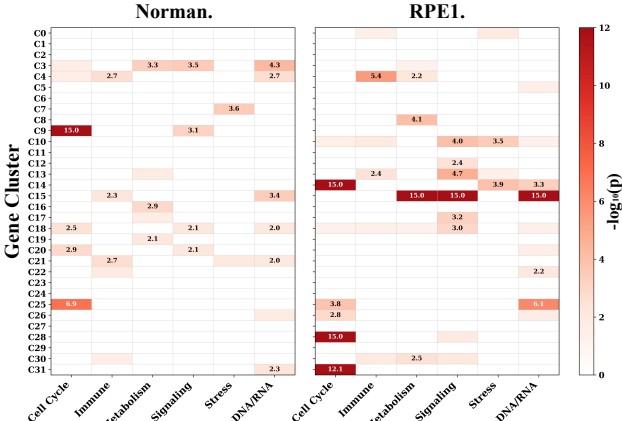

*Figure 3.* Functional enrichment analysis of GRC gene clusters across two datasets (§3.5).

## 3.5. Biological Analysis of Gene Clusters

**Functional Relevance of GRC Gene Modules.** To validate the biological significance of the partitions generated by GRC, we performed a post hoc external enrichment audit of

*Table 3.* Comparison of different methods across evaluation metrics on the RPE1 cell line(§3.3); Type abbreviations: Sta.=Statistical, Found.=Foundation, Gra.=Graph-based, Gen.=Generative.

| Type | Method | Public | $\rho\Delta\uparrow$ | $\rho\Delta^D\uparrow$ | ACC$\Delta\uparrow$ | ACC$\Delta^D\uparrow$ | DES↑ | PDS↑ | L2↓ | MSE↓ | MAE↓ |
|---|---|---|---|---|---|---|---|---|---|---|---|
| Sta. | Control | – | – | – | – | – | – | 0.5000 | 7.03 | 0.0171 | 0.0768 |
| | Linear | – | 0.2720 | 0.3630 | 0.5816 | 0.6919 | 0.2149 | 0.5318 | 14.13 | 0.0916 | 0.1491 |
| | Linear-scGPT (Ahlmann-Eltze et al., 2025) | *Nature Methods'25* | 0.3404 | 0.5438 | 0.5852 | 0.7681 | 0.2497 | 0.5338 | 7.60 | 0.0172 | 0.0890 |
| Found. | scGPT (Cui et al., 2024) | *Nature Methods'24* | 0.2840 | 0.5691 | 0.5882 | 0.7759 | 0.2980 | 0.5001 | 9.49 | 0.0246 | 0.1037 |
| | scFoundation (Hao et al., 2024) | *Nature Methods'24* | 0.2037 | 0.3632 | 0.5489 | 0.6908 | 0.2331 | 0.5098 | 7.07 | 0.0155 | 0.0812 |
| | GeneCompass (Yang et al., 2024) | *Cell Research'24* | 0.3113 | 0.4299 | 0.5758 | 0.7148 | 0.2873 | 0.5038 | 7.18 | 0.0162 | 0.0822 |
| | CellFM (Zeng et al., 2025) | *Cell Research'25* | 0.3060 | 0.4299 | 0.5733 | 0.7086 | 0.2819 | 0.5039 | 7.20 | 0.0163 | 0.0825 |
| Gra. | GEARS (Roohani et al., 2024) | *Nat. Biotechnol.'23* | 0.3083 | 0.4289 | 0.5752 | 0.7096 | 0.2820 | 0.5004 | 7.18 | 0.0162 | 0.0822 |
| | GraphVCI (Bereket & Karaletsos, 2023) | *ICLR'23* | 0.4294 | 0.5302 | 0.5589 | 0.6436 | 0.1863 | 0.5000 | 7.41 | 0.0157 | 0.0913 |
| Gen. | samsVAE (Wu et al., 2022) | *NeurIPS'23* | 0.4083 | 0.5204 | 0.5998 | 0.7565 | 0.3280 | 0.4984 | 6.86 | 0.0140 | 0.0780 |
| | STATE (Adduri et al., 2025) | *bioRxiv'25* | 0.3638 | 0.5298 | 0.6463 | 0.7891 | 0.3711 | 0.5011 | 8.38 | 0.0194 | 0.0759 |
| | CellFlow (Klein et al., 2025) | *bioRxiv'25* | 0.3878 | 0.4391 | 0.6226 | 0.7194 | 0.3114 | 0.5441 | 6.48 | 0.0135 | 0.0711 |
| | scBIG (Ours) | *Proposed* | **0.4875** | **0.5925** | **0.6471** | **0.8089** | **0.5145** | **0.5520** | **6.12** | **0.0118** | **0.0676** |

*Table 4.* Comparison of different methods across evaluation metrics on ComboSciPlex drug molecule perturbation(§3.4).

| Method | Public | $\rho\Delta\uparrow$ | $\rho\Delta^D\uparrow$ | MSE↓ | Discr. Cos↑ | E-Dist↓ | Wasserstein↓ |
|---|---|---|---|---|---|---|---|
| CPA (Lotfollahi et al., 2023) | Molecular Systems Biology '23 | 0.8387 | 0.9003 | 0.0652 | 0.8388 | 10.17 | 18.15 |
| scGPT (Cui et al., 2024) | *Nature Methods'24* | 0.8172 | 0.9280 | 0.0597 | 0.8030 | 14.12 | 17.46 |
| STATE (Adduri et al., 2025) | *bioRxiv'25* | 0.6233 | 0.7418 | 0.0904 | 0.6145 | 13.46 | 17.91 |
| CellFlow (Klein et al., 2025) | *bioRxiv'25* | 0.8276 | 0.9189 | 0.0461 | 0.8178 | 4.83 | 17.53 |
| scBIG (Ours) | *Proposed* | **0.8671** | **0.9321** | **0.0451** | **0.8612** | **4.74** | **16.76** |

the 32 gene modules using Reactome (Fabregat et al., 2018); Reactome annotations are not used during GRC module induction. As shown in Fig. 3, GRC effectively groups genes into functionally coherent modules across both datasets. In the Norman dataset, Cluster 9 and Cluster 3 are significantly enriched in *Cell Cycle* and *DNA/RNA-related* pathways, respectively. In the larger RPE1 dataset, these patterns become even more pronounced: Cluster 14 exhibits high specificity for *Cell Cycle*, while Cluster 15 serves as a functional hub for *Metabolism* and *Signaling*. These results confirm that GRC reliably identifies gene modules that mirror established biological hierarchies. Detailed enrichment results for specific subpathways are provided in §A.7.

**Perturbation-Specific Cross-Module Coordination.** This audit is necessarily partial: curated pathway priors are high-confidence but incomplete and context-agnostic; therefore, agreement with Reactome verifies that GRC recovers established functional structure, while perturbation-specific coordination provides complementary evidence beyond static annotations. As shown in Table 5, scBIG identifies stable cross-module coactivation patterns that are not directly specified by audited pathway priors. For example, under the CEBPE+ZC3HAV1 perturbation, M2 and M30 exhibit strong positive coactivation with a bootstrap positive rate of 1.00. Since M2 is chromatin/proteostasis-like and M30 is cytoskeleton/growth-like, this suggests a perturbation-specific coupling between differentiation/stress rewiring and downstream cytoskeletal-growth remodeling. These results indicate that GRC modules not only recover canonical biological programs, but also reveal condition-specific regulatory coordination beyond static pathway knowledge.

*Table 5.* Perturbation-specific cross-module coordination beyond audited pathway priors. Coactivation Score measures positive cross-module coactivation under the corresponding perturbation, and Bootstrap Positive Rate measures the stability of this positive coordination across bootstrap resampling (§3.5).

| Module Pair | Perturbation | Coactivation↑ | Bootstrap↑ |
|---|---|---|---|
| M2–M30 | CEBPE+ZC3HAV1 | 0.57 | 1.00 |
| M2–M8 | OSR2+UBASH3B | 0.52 | 0.98 |
| M2–M7 | OSR2+UBASH3B | 0.42 | 0.97 |

**Mechanistic Insights via Attention Shifts.** We further investigated whether GCAE leverages these modules to capture regulatory dependencies. Fig. 4 visualizes the attention shift ($\Delta$Attention) relative to the control condition for two representative perturbations in RPE1. For *NOL8* knockdown (a nucleolar protein essential for ribosome biogenesis), the model's attention to Cluster 8 (enriched in *mTORC1 Signaling* and *Metabolism*) surged by approximately **29%**, aligning with the known mechanism where impaired ribosome assembly triggers mTORC1 pathway dysregulation. Conversely, knockdown of *KCTD10*—a regulator of cullin-RING ligase complexes—induced an **18%** increase in attention toward Cluster 28 (*G2/M Checkpoint*), consistent with its role in cell-cycle progression. These case studies demonstrate that **scBIG** moves beyond black-box prediction to capture the underlying biological mechanisms of gene regulation.

### 3.6. Ablation Study

**Effectiveness of specific modules.** Table 6a summarizes the ablation results for different clustering and encoding strategies under the Norman additive setting. With the

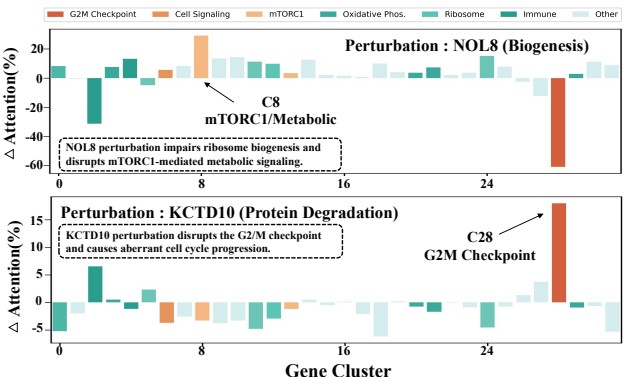

*Figure 4.* Differential attention (ΔAttention) across gene modules for two representative perturbations in RPE1 (§3.5).

GCAE encoder fixed, **GRC** clustering outperforms balanced K-means and the unordered baseline, yielding a **5.2%** and **1.5%** gain in $\rho\Delta$ and ACCΔ, respectively. This underscores GRC's superior ability to integrate functional similarities and PPI networks into precise module assignments. Furthermore, when GRC clustering is fixed, our GCAE significantly surpasses the standard Transformer, with $\rho\Delta$ and ACCΔ increasing by **1.8%** and **1.1%**. These results indicate that GCAE more effectively captures high-order inter-module dependencies.

Table 6b further ablates the clustering strategy and biological priors used in scBIG. Balanced OT clustering clearly outperforms unbalanced K-means by enforcing near-uniform chunk sizes, which assigns comparable attention capacity to each gene module and better preserves minority gene signals, improving $\rho_\Delta^D$ from 0.7765 to 0.8230. We also test Update by Pert, a perturbation-adaptive strategy that updates clusters based on perturbation prediction feedback. Its inferior performance suggests that stable GRC-induced modules offer a more reliable structural prior than repeatedly adapting clusters to individual perturbations. Moreover, removing PPI mainly hurts DEG-sensitive prediction, reducing $\rho_\Delta^D$ from 0.8230 to 0.7988, whereas removing GeneCompass embeddings primarily weakens global prediction, decreasing $\rho_\Delta$ from 0.8496 to 0.8326. These results show that GRC clustering, PPI structure, and GeneCompass embeddings provide complementary gains and jointly support the robustness of scBIG.

**Effectiveness of biological regularization and constraints.** As shown in Tables 6c and 6d, both biological losses are necessary and complementary. Using either $\mathcal{L}_{corr}$ or $\mathcal{L}_{pathway}$ alone yields moderate gains over the vanilla baseline, while jointly optimizing them achieves the best overall performance, with a **3.1%** improvement in $\rho\Delta$ and a **2.7%** gain in $\rho\Delta^D$. Notably, this trend holds across metrics, suggesting the two terms regularize distinct but complementary aspects of perturbation responses. Moreover, the weight sweep in Table 6d shows that over-emphasizing either term degrades

accuracy—especially on strongly DE genes—whereas the optimum occurs when $\lambda_{corr}$ and $\lambda_{pathway}$ are balanced. This clearly indicates that correlation alignment and pathway-informed transport regularize the model at different granularities, jointly improving module-level coordination and pathway-level fidelity.

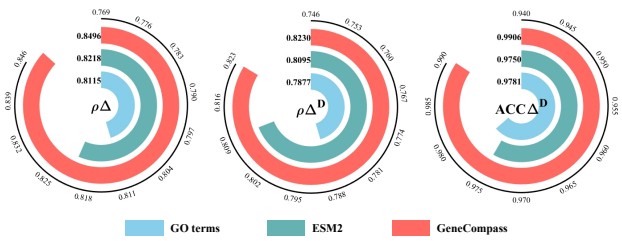

*Figure 5.* Comparison of different gene embeddings for clustering performance evaluation (§3.6).

**Impact of Gene Embeddings on Clustering and Perturbation Conditioning.** We further evaluate how different gene embeddings affect GRC clustering quality. As shown in Fig. 5, *GeneCompass* consistently outperforms both *GO terms* (Consortium, 2004) and *ESM2* (Lin et al., 2023), especially on $\rho\Delta$ and ACCΔ$^D$, suggesting that its context-aware semantics better support biologically coherent modules. Interestingly, when the same embeddings are used for *perturbation condition injection* (Table 6e), *ESM2* becomes stronger and achieves better downstream scores than *GeneCompass*. We attribute this reversal to their representation focus: *GeneCompass* emphasizes gene–context dependencies for module discovery, whereas *ESM2* encodes functional priors that transfer more reliably to unseen perturbations, improving out-of-distribution conditioning.

Finally, we report additional hyperparameter ablations and efficiency analyses in §A.1, and provide run-to-run standard deviations of evaluation metrics across multiple random seeds in §A.2.

## 4. Related Work

**Single-cell Foundation Models.** With the advancement of high-throughput sequencing technologies and transformer-based biomedical representation learning (Greco et al., 2026), large-scale single-cell scRNA-seq datasets have become widely available (Regev et al., 2017; Zheng et al., 2017). Recent foundation models, such as scGPT (Cui et al., 2024), scFoundation (Hao et al., 2024), GeneCompass (Yang et al., 2024), and CellFM (Zeng et al., 2025) leverage pretraining on massive single-cell corpora to learn gene embeddings that capture rich co-expression patterns and contextual semantics, enabling strong performance on downstream tasks including batch correction and cell annotation. However, most of these models operate on a *flat* gene-token representation without an explicit program/module

*Table 6.* A set of ablative experiments on the Norman additive split (§3.6).

**(a)** Clustering and encoding strategies (§3.6).

| Cluster | | Encoding | | Norman Additive | | | |
|---|---|---|---|---|---|---|---|
| K-means | GRC | Normal | GCAE | $\rho\Delta$ | $\rho\Delta^{\mathrm{D}}$ | ACC$\Delta$ | ACC$\Delta^{\mathrm{D}}$ |
| | | ✓ | | 0.7781 | 0.7885 | 0.8849 | 0.9609 |
| | | | ✓ | 0.8078 | 0.8109 | 0.9008 | 0.9625 |
| ✓ | | ✓ | | 0.8239 | 0.8052 | 0.9059 | 0.9781 |
| ✓ | | | ✓ | 0.8358 | 0.8190 | 0.9161 | 0.9813 |
| | ✓ | ✓ | | 0.8348 | 0.8226 | 0.9097 | 0.9859 |
| | ✓ | | ✓ | **0.8496** | **0.8230** | **0.9197** | **0.9906** |

**(b)** Comparison of clustering strategies (§3.6).

| Configuration | $\rho\Delta$ | $\rho\Delta^{\mathrm{D}}$ | ACC$\Delta$ | ACC$\Delta^{\mathrm{D}}$ |
|---|---|---|---|---|
| K-means Unbalanced | 0.8249 | 0.7765 | 0.9163 | 0.9719 |
| Soft Cluster ($\tau=0.5$) | 0.8408 | 0.7923 | **0.9203** | 0.9797 |
| Soft Cluster ($\tau=0.9$) | 0.8411 | 0.7904 | 0.9200 | 0.9797 |
| Update by Pert | 0.8401 | 0.8065 | 0.9081 | 0.9867 |
| Ours w/o any prior | 0.8235 | 0.8070 | 0.8874 | 0.9656 |
| Ours w/o GC emb | 0.8326 | **0.8234** | 0.9099 | **0.9906** |
| Ours w/o PPI | 0.8396 | 0.7988 | 0.9190 | 0.9813 |
| Ours | **0.8496** | 0.8230 | 0.9197 | **0.9906** |

**(c)** Effectiveness of biological loss components (§3.6).

| $\mathcal{L}_{\mathrm{corr}}$ | $\mathcal{L}_{\mathrm{pathway}}$ | $\rho\Delta$ | $\rho\Delta^{\mathrm{D}}$ | ACC$\Delta$ | ACC$\Delta^{\mathrm{D}}$ |
|---|---|---|---|---|---|
| | | 0.8242 | 0.8014 | 0.9076 | 0.9703 |
| ✓ | | 0.8306 | 0.8146 | 0.9109 | 0.9859 |
| | ✓ | 0.8361 | 0.8050 | 0.9164 | **0.9922** |
| ✓ | ✓ | **0.8496** | **0.8230** | **0.9197** | 0.9906 |

**(d)** Impact of loss weights (§3.6).

| $\lambda_{\mathrm{corr}}$ | $\lambda_{\mathrm{pathway}}$ | $\rho\Delta$ | $\rho\Delta^{\mathrm{D}}$ | ACC$\Delta$ | ACC$\Delta^{\mathrm{D}}$ |
|---|---|---|---|---|---|
| 0.1 | 1.0 | 0.8437 | 0.7987 | 0.9179 | 0.9859 |
| 1.0 | 0.1 | 0.8393 | 0.8162 | 0.9119 | 0.9891 |
| 0.1 | 0.1 | 0.8496 | 0.8230 | 0.9197 | 0.9906 |

**(e)** Comparison of gene embeddings for perturbed gene conditions (§3.6).

| Condition Embedding | $\rho\Delta$ | $\rho\Delta^{\mathrm{D}}$ | ACC$\Delta$ | ACC$\Delta^{\mathrm{D}}$ |
|---|---|---|---|---|
| GeneCompass | 0.8397 | 0.8182 | 0.9164 | 0.9750 |
| ESM2 | **0.8496** | **0.8230** | **0.9197** | **0.9906** |

scaffold, and are not optimized to preserve coordinated program-level structure under perturbations. This design choice may limit their generalization in perturbation response prediction, particularly for unseen perturbations.

**Generative Perturbation Modeling.** Recent generative frameworks (Lotfollahi et al., 2019; Wu et al., 2022; Lotfollahi et al., 2023; Adduri et al., 2025), together with broader progress in in-context conditional generation (Zhang et al., 2026; Song et al., 2026), have shown strong performance in modeling structured outputs and extrapolating to unseen conditions. Unlasting (Chi et al., 2025a) and CellFlow (Klein et al., 2025) further improve distributional consistency by learning continuous transitions between control and perturbed states in a latent space, capturing smooth trajectories of gene-expression change. However, these models typically operate on flat gene-token representations or weakly structured latent bottlenecks and do not explicitly enforce *module-level coordination*. As a result, they may not preserve cross-program structure (*e.g.*, module correlation patterns and pathway-level organization), which is critical for robust generalization under combinatorial and out-of-distribution perturbations.

**Incorporating Biological Priors.** A growing body of work enhances perturbation modeling by incorporating biological priors into the architecture (Chi et al., 2025b). GEARS (Roohani et al., 2024) integrates Gene Ontology (GO) (Consortium, 2004) and interaction-derived graphs via GNNs (Scarselli et al., 2008) to propagate signals across genes, while CellOracle (Kamimoto et al., 2023) leverages gene regulatory networks (GRNs) (Karlebach & Shamir, 2008) to simulate responses. These approaches improve interpretability by grounding predictions in structured biological knowledge. However, the effectiveness of such models often scales with the quality of the underlying graphs, which may be subject to incomplete coverage, noise, or the inability to fully reflect dynamic, perturbation-induced regulatory

rewiring (Milano et al., 2022). Consequently, there remains a critical need for strategies that can effectively bridge these static priors with broader, context-aware biological insights.

Compared with fixed graph propagation methods, our approach introduces a shift toward knowledge-augmented modularity (Quan et al., 2024). Instead of relying solely on potentially incomplete static neighborhoods, scBIG leverages the rich contextual semantics from single-cell foundation models to induce functional gene programs. This design choice enables a more comprehensive coverage of gene relationships that may be absent in traditional graphs, allowing the model to capture high-order, cross-program coordination. By fusing molecular priors with the deep semantic embeddings of foundation models, scBIG moves beyond the limitations of isolated gene tokens and static local propagation to a more robust, module-aware framework.

## 5. Conclusion

Regarding the challenge of generalizing perturbation prediction to unseen genes and combinatorial interventions, we present scBIG, a module-inductive generative framework for modeling transcriptional responses with high structural fidelity. Compared to approaches that operate on flat gene-wise representations or rely on static and incomplete biological graphs, scBIG has merits in: **i)** *knowledge-augmented module induction* that adaptively constructs biologically coherent gene programs by fusing foundation-model semantics with high-confidence PPI priors; **ii)** *module-aware generative modeling* that captures inter-program dependencies through a Gene-Cluster-Aware Encoder within a conditional flow-matching backbone; **iii)** *structure-aware objectives* that preserve coordinated program-level behavior by enforcing module correlation consistency and pathway-level alignment in generated profiles. Extensive experiments on two widely used benchmarks confirm the effectiveness and robustness of scBIG across diverse perturbation settings.

## Acknowledgments

This work is supported by the National Science and Technology Major Project (2023ZD0120801) and the Scientific Computing Intelligent Solver project, funded by the Fundamental Research Funds for the Central Universities (226-2025-00080).

## Impact Statement

This study strictly utilizes publicly available single-cell transcriptomic datasets in full compliance with their respective Data Use Agreements. Our analysis involves no personally identifiable information (PII) or sensitive patient metadata. As a computational framework for in silico hypothesis generation, our model is intended for research purposes and does not constitute a clinical diagnostic tool. While it holds potential for accelerating drug discovery, all predictions require rigorous in vitro and in vivo validation before any therapeutic application. We advocate for the responsible use of generative AI in biology and caution against interpreting model outputs without proper biological context.

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

# A. Appendix

The appendix is organized as follows:

- §A.1 offers more ablation experiments of scBIG, including analysis of key hyperparameters and efficiency.

- §A.2 shows the robustness of evaluation on the Norman (Norman et al., 2019) dataset.

- §A.3 provides additional quantitative and qualitative analyses on the Norman (Norman et al., 2019) dataset.

- §A.4 provides more implementation details of scBIG.

- §A.5 evaluates cross-cell-line transfer from K562 to RPE1.

- §A.6 offers the detail of baseline methods.

- §A.7 analysis the pathway of gene clusters.

- §A.8 discusses the limitations and future work.

## A.1. Ablation Study

### A.1.1. IMPACT OF STRUCTURAL HYPERPARAMETERS.

Fig. 6 illustrates the model's sensitivity to the number of clusters ($K$) and inducing points ($M$) under the Norman additive setting. We observe that a moderate structural constraint provides the optimal balance between model complexity and predictive fidelity. Specifically, reducing $K$ from 128 to 32 consistently improves performance across all metrics, with the most robust results achieved at $K = 32$ and $M = 8$. These findings suggest that excessively fine-grained clustering may introduce noise, whereas a well-calibrated $K$ effectively captures the functional modularity of gene regulation.

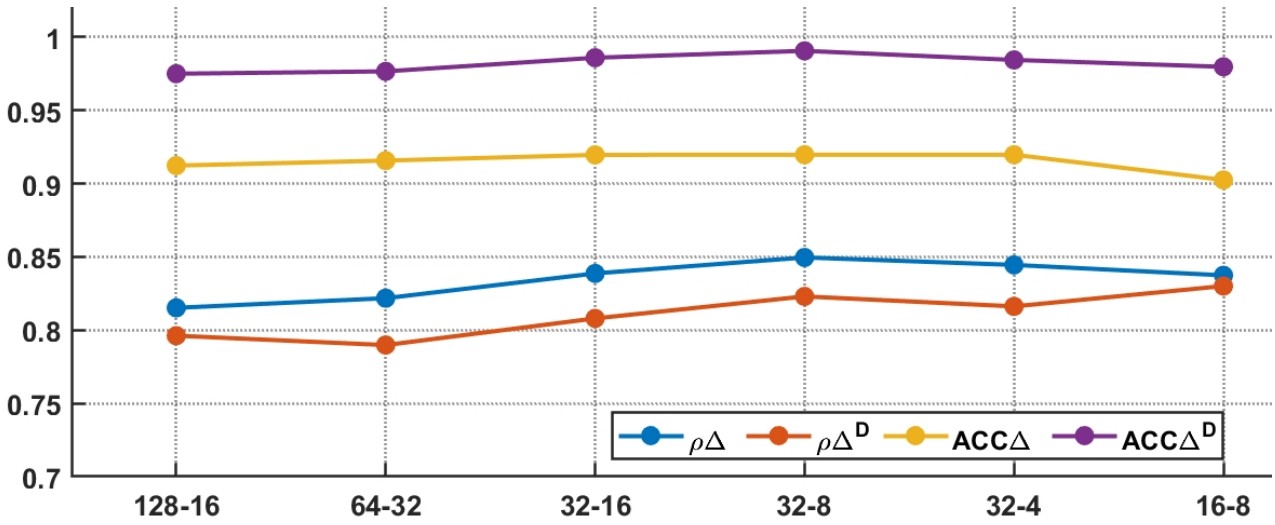

*Figure 6.* Performance of different configurations (cluster number and inducing point dimension) on various metrics (§A.1.1).

### A.1.2. EFFICIENCY ANALYSIS OF SCBIG

Table 7 presents the computational overhead incurred by our proposed biological consistency losses. We benchmark the per-step training time with a batch size of 1024 on a single NVIDIA RTX 3090 GPU. The baseline model, without biological losses, requires $83.17$ ms per training step. Incorporating the cluster correlation loss $\mathcal{L}_{\text{corr}}$ results in $81.94$ ms, while the pathway optimal transport loss $\mathcal{L}_{\text{pathway}}$ (incorporating 969 Reactome pathways and 100 Sinkhorn iterations) increases the step time to $87.91$ ms. Integrating both losses yields $87.44$ ms per step.

Notably, the addition of the cluster correlation loss alone shows a marginal reduction in training time ($-1.23$ ms) compared to the baseline. This phenomenon is likely attributed to two factors: (1) the computation of $\mathcal{L}_{corr}$ is highly efficient, involving only gene-to-cluster aggregation and a small $32 \times 32$ correlation matrix; and (2) JAX's XLA compiler may identify more efficient execution strategies when optimizing the augmented computational graph, as structural changes can trigger different kernel fusion opportunities. This observed variance falls within the expected measurement noise (std $\approx 1$ ms).

In contrast, the pathway OT loss introduces a measurable overhead of approximately $4.7$ ms ($+5.7\%$), primarily due to the iterative Sinkhorn algorithm. The total overhead for both biological losses remains modest at approximately $5.1\%$, demonstrating that our biologically-informed objectives can be incorporated with minimal computational penalty.

*Table 7.* Computational overhead of biological consistency losses(§A.1.2).

| $\mathcal{L}_{corr}$ | $\mathcal{L}_{pathway}$ | Batch Size | Time/Step (ms) |
|---|---|---|---|
| | | 1024 | 83.17 |
| ✓ | | 1024 | 81.94 |
| | ✓ | 1024 | 87.91 |
| ✓ | ✓ | 1024 | 87.44 |

Table 8 analyzes the computational overhead of pretrained encoder-decoder relative to the number of gene clusters $K$, which determines the granularity of our Gene-Cluster Aware Encoder (GCAE). With $K = 1$ (equivalent to global attention over all 2,051 genes), the model achieves a training time of $12.21$ ms per step with a batch size of 256. As $K$ increases, the cost grows due to the increased frequency of attention operations: $K = 32$ requires $29.91$ ms, while $K = 512$ increases to $198.6$ ms. In the extreme case of $K = 2,051$ (per-gene attention), memory constraints necessitate reducing the batch size to 128, resulting in $394.16$ ms per step.

These results validate that our GCAE design with a moderate $K$ (*e.g.*, $K = 32$) provides an optimal trade-off between computational tractability and functional resolution. It maintains efficient throughput while enabling structured modeling of complex gene-gene interactions.

*Table 8.* Computational overhead of different gene cluster numbers(§A.1.2).

| $K$ | Batch Size | Time/Step (ms) |
|---|---|---|
| 1 | 256 | 12.2 |
| 16 | 256 | 22.4 |
| 32 | 256 | 29.9 |
| 128 | 256 | 65.2 |
| 512 | 256 | 198.6 |
| 2051 | 128 | 394.2 |

Table 9 further reports scalability on a single RTX 4090 24GB GPU. The default 2k-HVG Norman setting uses modest $K/M$; larger gene sets remain trainable by increasing $K$ and $M$ with the gene set size while reducing batch size when needed.

### A.1.3. POOLING STRATEGY

Since inter-module interaction has already been modeled through the Perceiver bottleneck, we use mean pooling as a simple and stable readout. Additional set-level pooling introduces unnecessary complexity and performs worse in our ablations, as shown in Table 10.

### A.2. Robustness on the Norman dataset

To assess the stability of scBIG and its sensitivity to stochastic initialization, we conducted a rigorous robustness analysis across five independent trials. For each trial, the model was trained and evaluated using distinct random seeds, ensuring that the results are not artifacts of a specific initialization.

As summarized in Table 11 and Table 12, we report the standard deviation for each evaluation metric. The observed standard deviations remain consistently low across all tasks and data partitions, demonstrating that our module-inductive framework and flow-matching backbone are numerically stable.

### A.3. Additional Quantitative and Qualitative Analyses on Norman

### A.3.1. DISTRIBUTION-LEVEL QUANTITATIVE COMPARISON

To further assess whether predicted cells preserve the full perturbation-conditional distribution beyond mean-level accuracy, we evaluate three distribution-level metrics on the Norman additive split: discriminative cosine similarity, energy distance,

*Table 9.* Scalability on a single NVIDIA RTX 4090 24GB GPU. The Norman default uses 2,051 HVGs with $K = 32$ and $M = 8$(§A.1.2).

| Setting | $K$ | $M$ | Latent | HVG | $\rho\Delta$ | $\rho\Delta^{\mathrm{D}}$ | ACC$\Delta$ | ACC$\Delta^{\mathrm{D}}$ | Batch 1 | Mem. 1 | Batch 2 | Mem. 2 | Step ms |
|---|---|---|---|---|---|---|---|---|---|---|---|---|---|
| Norman default | 32 | 8 | 50 | 2051 | 0.8496 | 0.8230 | 0.9197 | 0.9906 | 256 | 4G | 1024 | 23G | 87.44 |
| 5k HVGs | 32 | 8 | 50 | 5018 | 0.7921 | 0.8591 | 0.6933 | 0.9719 | 256 | 8G | 1024 | 23G | 43.42 |
| 5k HVGs | 64 | 16 | 64 | 5018 | 0.8040 | 0.8691 | 0.7079 | 0.9594 | 256 | 8G | 1024 | 24G | 53.65 |
| 10k HVGs | 32 | 8 | 50 | 10000 | 0.7683 | 0.8428 | 0.6571 | 0.9656 | 128 | 17G | 256 | 19G | 27.64 |
| 10k HVGs | 128 | 32 | 96 | 10000 | 0.7704 | 0.8516 | 0.6758 | 0.9562 | 128 | 17G | 256 | 22G | 30.64 |

*Table 10.* Comparison of pooling strategies on the Norman additive split(§A.1.3).

| **Strategy** | $\rho\Delta$ | $\rho\Delta^{\mathrm{D}}$ | **ACC$\Delta$** | **ACC$\Delta^{\mathrm{D}}$** |
|---|---|---|---|---|
| Avg pool | **0.8496** | **0.8230** | **0.9197** | **0.9906** |
| Set Transformer | 0.8266 | 0.7971 | 0.9142 | 0.9812 |
| Attention | 0.8363 | 0.8124 | 0.9181 | 0.9828 |

and Sinkhorn-approximated Wasserstein distance. As shown in Table 13, scBIG achieves the best performance across all three metrics, improving over CellFlow from 0.6397 to 0.7264 in discriminative cosine similarity and reducing E-Dist and Wasserstein distance from 1.1979 to 0.8383 and from 10.7579 to 9.7842, respectively. These results indicate that the generated cells not only match perturbation directions more accurately but also better recover the cell-level response distribution.

### A.3.2. STATISTICAL SIGNIFICANCE

We report mean $\pm$ standard error across perturbations and paired Wilcoxon tests against CellFlow. As shown in Table 14, the gains remain significant across perturbations rather than arising only from marginal averages.

### A.3.3. CLUSTERING AND ENCODER UNCERTAINTY

Table 15 adds 95% confidence intervals for the clustering and encoder comparison. The result should be interpreted as a stable ordering and complementary gain from biologically informed clustering and GCAE, rather than a dramatic step-change claim.

### A.3.4. QUALITATIVE DISTRIBUTION VISUALIZATION

We further compare scBIG with CellFlow through UMAP density visualizations on representative unseen combinatorial perturbations from the Norman additive split. As shown in Figure 7, scBIG produces density contours that better overlap with the ground-truth cell distributions, while CellFlow shows visible mismatches in boundary regions. This qualitative comparison further supports the distributional fidelity of scBIG.

## A.4. Experiments Setup

### A.4.1. DEFAULT SETTINGS

**Implementation Details.** The scBIG framework was trained on a single NVIDIA RTX 3090 GPU using the JAX (Bradbury et al., 2018) deep learning framework with the Flax neural network library. Training consists of two stages: (1) encoder-decoder pretraining and (2) flow matching with biological consistency fine-tuning. The overall pipeline is summarized in Algorithms 1 to 3.

Before training, we obtain gene modules $\mathcal{C}$ via Gene-Relation Clustering (GRC), as described in Algorithm 1.

**Stage 1: Encoder-Decoder Pretraining.** The GCAE encoder and decoder are jointly pretrained for 100 epochs using the Adam (Kingma, 2014) optimizer with a learning rate of $10^{-4}$ and batch size of 256. The encoder employs a Perceiver-style induced attention mechanism with 8 inducing points per gene module, reducing computational complexity from $O(G^2)$ to $O(G \cdot M)$ where $G$ is the number of genes and $M$ is the number of inducing points. The Stage I pretraining procedure is detailed in Algorithm 2.

**Stage 2: Joint Flow Matching with Biological Consistency.** In the second stage, the flow matching loss and biological consistency losses are **jointly optimized at every training step**. The conditional flow network is trained for 500,000 iterations with a batch size of 1024 (accumulated over 20 gradient steps). We use the Adam optimizer with gradient clipping (max norm 1.0) and a learning rate of $5 \times 10^{-5}$. We do not assume one-to-one paired control and perturbed cells; instead,

*Table 11.* Standard deviation across five seeds on the Norman additive split(§A.2).

| Method | Public | $\rho\Delta$ | $\rho\Delta^{\mathrm{D}}$ | ACC$\Delta$ | ACC$\Delta^{\mathrm{D}}$ | DES | PDS | L2 | MSE | MAE |
|---|---|---|---|---|---|---|---|---|---|---|
| GEARS | *Nat. Biotechnol.'23* | 0.0055 | 0.0151 | 0.0031 | 0.0080 | 0.0081 | 0.0048 | 0.0649 | 0.0006 | 0.0014 |
| scGPT | *Nature Methods'24* | 0.0006 | 0.0045 | 0.0010 | 0.0010 | 0.0084 | 0.0026 | 0.0197 | 0.0002 | 0.0004 |
| scFoundation | *Nature Methods'24* | 0.0009 | 0.0031 | 0.0004 | 0.0036 | 0.0067 | 0.0029 | 0.0044 | 0.0001 | 0.0002 |
| CellFlow | *bioRxiv'25* | 0.0035 | 0.0146 | 0.0013 | 0.0213 | 0.0033 | 0.0074 | 0.0636 | 0.0004 | 0.0014 |
| Ours | *Proposed* | 0.0026 | 0.0038 | 0.0023 | 0.0029 | 0.0015 | 0.0096 | 0.0411 | 0.0002 | 0.0009 |

*Table 12.* Standard deviation across five seeds on the Norman holdout split(§A.2).

| Method | Public | $\rho\Delta$ | $\rho\Delta^{\mathrm{D}}$ | ACC$\Delta$ | ACC$\Delta^{\mathrm{D}}$ | DES | PDS | L2 | MSE | MAE |
|---|---|---|---|---|---|---|---|---|---|---|
| GEARS | *Nat. Biotechnol.'23* | 0.0072 | 0.0122 | 0.0042 | 0.0091 | 0.0158 | 0.0023 | 0.0227 | 0.0001 | 0.0005 |
| scGPT | *Nature Methods'24* | 0.0008 | 0.0027 | 0.0003 | 0.0031 | 0.0155 | 0.0006 | 0.0039 | 0.0001 | 0.0001 |
| scFoundation | *Nature Methods'24* | 0.0072 | 0.0122 | 0.0042 | 0.0091 | 0.0158 | 0.0023 | 0.0227 | 0.0000 | 0.0005 |
| CellFlow | *bioRxiv'25* | 0.0026 | 0.0098 | 0.0013 | 0.0038 | 0.0042 | 0.0092 | 0.0053 | 0.0000 | 0.0001 |
| Ours | *Proposed* | 0.0016 | 0.0036 | 0.0021 | 0.0014 | 0.0042 | 0.0026 | 0.0105 | 0.0000 | 0.0002 |

control and perturbation cells are sampled separately and mini-batch OT is used to construct transport pairs in the learned latent space. The Stage II joint optimization procedure is detailed in Algorithm 3.

To solve the Ordinary Differential Equations (ODEs) during both training (for consistency losses) and inference, we follow the configuration of *CellFlow* (Klein et al., 2025).

### A.4.2. DATASET DETAILS

Transcriptomic profiles were sourced from the scPerturb resource (Peidli et al., 2024). To ensure robust downstream analysis, we implemented a rigorous preprocessing pipeline following the established `Scanpy` framework (Wolf et al., 2018):

- **Quality Control (QC):** We enforced stringent filtering criteria to eliminate low-quality cells and stochastic noise, specifically retaining cells with a minimum of 1,000 detected genes and genes expressed in at least 50 individual cells.

- **Feature Engineering:** For each dataset, we prioritized the top 2,000 highly variable genes (HVGs) to capture the most informative transcriptional variations. Crucially, all perturbed genes were manually appended to the feature set to ensure the inclusion of primary regulatory targets.

- **Transformation:** Raw counts underwent library-size normalization and log-transformation ($X = \ln(\mathrm{CPM} + 1)$) to stabilize variance and mitigate sequencing depth artifacts.

- **Differential Expression (DE) Profiling:** To characterize the transcriptomic impact of each perturbation, we identified differentially expressed genes (DEGs) via the Wilcoxon rank-sum test. Genes with Benjamini-Hochberg adjusted p-values below 0.05 were designated as DEGs, providing a ground-truth reference for functional response evaluation.

The Norman dataset (Norman et al., 2019) serves as a pivotal benchmark for modeling high-dimensional transcriptional responses to combinatorial genetic perturbations. This study employed CRISPR activation (CRISPRa) to systematically upregulate target genes within the human K562 leukemia cell line. The resulting Perturb-seq atlas comprises transcriptomic profiles of approximately 91,000 single cells, encompassing 105 single-gene and 131 dual-gene activations. Due to its coverage of complex genetic interactions—including synergistic, epistatic, and redundant effects—this dataset poses a significant challenge for generative models aiming to generalize from individual components to unseen combinatorial states.

To rigorously evaluate model generalization, we designed two distinct experimental settings, summarized in Fig. 8. The first, the additive setting, is designed to assess compositional generalization, where a model is trained on single-gene perturbations and must predict the effects of unseen dual-gene combinations. The second, the more challenging holdout setting, evaluates zero-shot generalization. In this scenario, a subset of genes is entirely held out from training, and the model must predict the perturbation outcomes for these unseen genes, both individually and in novel combinations. Together, these settings provide a comprehensive benchmark of a model's ability to extrapolate beyond its training data.

The Replogle2022_rpe1 dataset (Replogle et al., 2022) is an essential-gene Perturb-seq screen in the hTERT-immortalized retinal pigment epithelial (RPE1) cell line, generated by Replogle et al. (2022). Using CRISPR interference (CRISPRi), the study systematically silences common essential genes (on the order of ~2.4k targets) and profiles transcriptomic responses at single-cell resolution, comprising on the order of hundreds of thousands of cells. As a large-scale, high-quality Perturb-seq

*Table 13.* Distribution-level comparison on the Norman additive split. Metrics are macro-averaged over perturbations. Higher is better for Discr. Cos, while lower is better for E-Dist and Wasserstein(§A.3.1).

| Type | Method | Discr. Cos↑ | E-Dist↓ | Wasserstein↓ |
|------|--------|-------------|---------|--------------|
| Sta. | Control | -0.3009 | 13.5125 | 16.5615 |
|      | Linear | 0.3967 | 11.1376 | 15.3741 |
| Found. | scFoundation (Hao et al., 2024) | 0.6218 | 1.3710 | 11.2452 |
|        | GeneCompass (Yang et al., 2024) | 0.5623 | 1.3059 | 11.1429 |
| Gra. | GEARS (Roohani et al., 2024) | 0.5871 | 1.2377 | 11.0829 |
| Gen. | CellFlow (Klein et al., 2025) | 0.6397 | 1.1979 | 10.7579 |
|      | scBIG (Ours) | **0.7264** | **0.8383** | **9.7842** |

*Table 14.* Paired comparison against CellFlow on the Norman additive split. We report mean $\pm$ standard error across perturbations and paired Wilcoxon p-values(§A.3.2).

| Method | $\rho\Delta\uparrow$ | $\rho\Delta^{\mathrm{D}}\uparrow$ | $\mathrm{ACC}\Delta\uparrow$ | $\mathrm{ACC}\Delta^{\mathrm{D}}\uparrow$ |
|--------|----------------------|-----------------------------------|------------------------------|--------------------------------------------|
| CellFlow (Klein et al., 2025) | $0.788 \pm 0.002$ | $0.730 \pm 0.006$ | $0.913 \pm 0.002$ | $0.940 \pm 0.005$ |
| scBIG (Ours) | $\mathbf{0.846 \pm 0.005}$ | $\mathbf{0.814 \pm 0.013}$ | $\mathbf{0.917 \pm 0.004}$ | $\mathbf{0.990 \pm 0.001}$ |
| Paired Wilcoxon $p$ | $4.4\times10^{-5}$ | $0.011$ | $0.043$ | $0.002$ |

benchmark, it provides a rigorous testbed for generative models to predict cellular states under diverse essential gene knockdowns and to assess scalability and generalization in complex regulatory settings.

The ComboSciPlex dataset (Lotfollahi et al., 2023; Yu et al., 2026) provides a complementary drug-perturbation benchmark, where single-cell transcriptional profiles are measured after chemical treatments across diverse compounds and dosages. We use the top 2,000 highly variable genes (HVG2000) and evaluate the model under a held-out drug perturbation protocol, testing whether scBIG generalizes beyond genetic interventions to small-molecule response prediction.

**Dataset Partitioning.** We partitioned the filtered perturbations into training, validation, and testing sets. For Norman and ComboSciPlex, we follow the experimental split protocol of scDFM (Yu et al., 2026); for Replogle2022_rpe1, we follow the default protocol established in GEARS (Roohani et al., 2024). The training set was utilized for model fitting, while the validation set was employed for hyperparameter tuning and model selection. Finally, the testing set was reserved for an unbiased evaluation of the final model's performance. For a detailed breakdown of the genetic-perturbation splits, please refer to Table 17.

### A.4.3. REPRODUCIBILITY STATEMENT

Our code is provided in the supplementary materials, including the complete implementation framework. To ensure transparency and reproducibility, we will release all datasets, codebases, and model checkpoints, accompanied by comprehensive documentation.

### A.4.4. METRIC DEFINITIONS

To comprehensively evaluate the point-prediction accuracy and distributional fidelity of our method, we employ a diverse set of metrics established in prior single-cell perturbation modeling studies, including those utilized in the Virtual Cell Challenge (Roohani et al., 2025). The evaluation protocol for the majority of these metrics remains consistent with the standard pipeline defined in `cell-evel` library.

**Point-prediction Accuracy:** These metrics quantify the numerical deviation between the predicted and ground-truth gene expression profiles:

- **MSE & MAE:** Mean Squared Error and Mean Absolute Error measure the cell-level or pseudobulk-level residual magnitudes.

$$\mathrm{MSE} = \frac{1}{NG} \sum_{i=1}^{N} \sum_{j=1}^{G} (X_{ij} - \hat{X}_{ij})^2, \quad \mathrm{MAE} = \frac{1}{NG} \sum_{i=1}^{N} \sum_{j=1}^{G} |X_{ij} - \hat{X}_{ij}| \tag{10}$$

  where $N$ and $G$ denote the number of cells and genes, respectively.

*Table 15.* Clustering and encoder comparison with 95% confidence intervals on the Norman additive split(§A.3.3).

| Clustering | Encoder | $\rho\Delta \uparrow$ | 95% CI |
|---|---|---|---|
| No clustering | GCAE | 0.808 | [0.754, 0.882] |
| K-means | Normal | 0.824 | [0.769, 0.886] |
| K-means | GCAE | 0.836 | [0.793, 0.894] |
| GRC | Normal | 0.835 | [0.788, 0.895] |
| GRC | GCAE | **0.850** | **[0.799, 0.900]** |

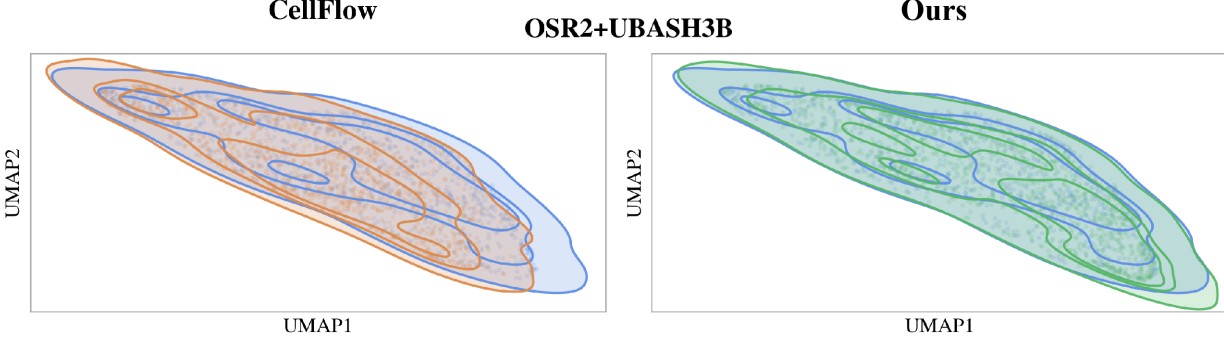

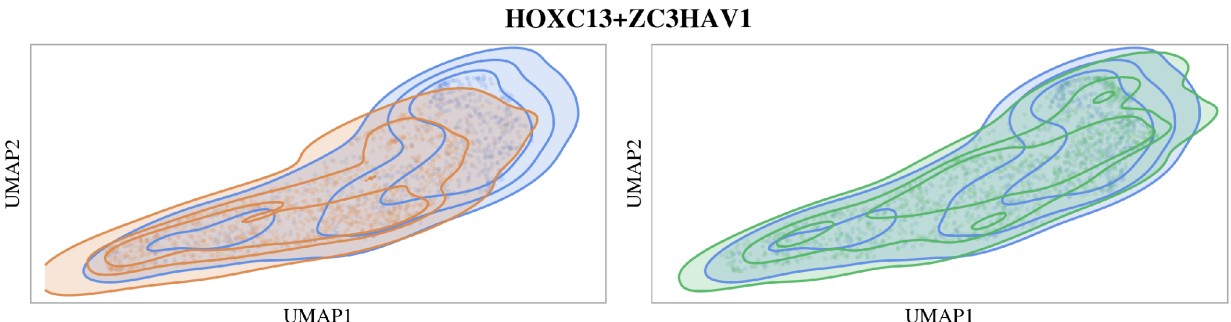

*Figure 7.* UMAP density visualizations on representative Norman additive perturbations. Ground truth is shown in blue, CellFlow in orange, and scBIG in green. Red circles indicate regions where CellFlow deviates from the ground-truth distribution, while scBIG shows better density overlap(§A.3.4).

- **L2 Distance** ($L_{2,\mathbf{mean}}$): Measures the Euclidean distance between the predicted and observed mean expression vectors across all perturbations $p \in \mathcal{P}$.

$$L_{2,\text{mean}} = \frac{1}{|\mathcal{P}|} \sum_{p \in \mathcal{P}} \|\hat{\mathbf{y}}_p - \mathbf{y}_p\|_2 \tag{11}$$

**Biological Response Fidelity**  To evaluate the model's capacity to capture complex transcriptional dynamics and regulatory directions, we utilize the following metrics.

- **Pearson $\Delta$ ($\rho\Delta$) & Pearson $\Delta$ DEG ($\rho\Delta^{\mathrm{D}}$):** These metrics measure the linear correlation between predicted and observed perturbation-induced transcriptional shifts, defined as $\Delta\mathbf{y} = \mathbf{y}_p - \bar{\mathbf{y}}_{\text{ctrl}}$.

$$\rho\Delta = \frac{1}{|\mathcal{P}|} \sum_{p \in \mathcal{P}} \text{Corr} \left( \hat{\mathbf{y}}_p - \bar{\mathbf{y}}_{\text{ctrl}}, \mathbf{y}_p - \bar{\mathbf{y}}_{\text{ctrl}} \right) \tag{12}$$

Specifically, $\rho\Delta^{\mathrm{D}}$ restricts this evaluation to the **top 20 highly-responsive genes** (DEGs) ranked by their absolute mean difference, providing a focused assessment of the model's ability to recover primary regulatory signals while mitigating stochastic noise.

*Table 16.* Experiment Configurations(§A.4.1).

| Category | Hyperparameter | Value |
|---|---|---|
| Model Settings | Latent dimension $D$ | 50 |
| | Embedding dimension | 256 |
| | Encoder layers | 3 |
| | Attention heads | 8 |
| | Inducing points $M$ | 8 |
| | Gene clusters $K$ | 32 |
| | Flow hidden layers | 2 |
| | Flow hidden dimension | 1024 |
| | Flow dropout | 0.1 |
| | Condition embedding dim | 256 |
| Stage 1 Pretrain | Epochs | 100 |
| | Batch size | 256 |
| | Learning rate | $10^{-4}$ |
| | Optimizer | Adam |
| Stage 2 Flow | Iterations | 500,000 |
| | Batch size $B$ | 1024 |
| | Gradient accumulation | 20 |
| | Gradient clip norm | 1.0 |
| | Optimizer | Adam |
| | Learning rate | $5 \times 10^{-5}$ |
| | $\lambda_{\mathrm{corr}}$ | 0.1 |
| | $\lambda_{\mathrm{pathway}}$ | 0.1 |

*Table 17.* Summary Statistics of Different Datasets(§A.4.2)

| Metric | Norman Additive (Norman et al., 2019) | Norman Holdout (Norman et al., 2019) | Replogle2022_rpe1 (Replogle et al., 2022) |
|---|---|---|---|
| Number train/val/test | 89926/8311/10990 | 92213/7905/11327 | 109929/11191/41999 |
| Condition of train/val/test | 167/32/32 | 190/21/26 | 1037/115/384 |
| Test Description | Only double | Single & Double | Only single |
| UMI Count | 2051 | 2051 | 3754 |
| Cell Type | K562 | K562 | RPE1 |

- **Directional Accuracy (ACC$\Delta$ & ACC$\Delta^{\mathrm{D}}$):** These metrics quantify the model's precision in predicting the direction of gene expression changes (up-regulation vs. down-regulation) relative to the control state.

$$\mathrm{ACC}\Delta = \frac{1}{|\mathcal{P}|G} \sum_{p \in \mathcal{P}} \sum_{g=1}^{G} \mathbb{I}\left[\mathrm{sgn}(\hat{y}_{pg} - \bar{y}_{g,\mathrm{ctrl}}) = \mathrm{sgn}(y_{pg} - \bar{y}_{g,\mathrm{ctrl}})\right] \tag{13}$$

Analogous to the correlation metrics, ACC$\Delta^{\mathrm{D}}$ focuses exclusively on the most significantly perturbed gene subset (top 20 DEGs) to verify the robustness of predicted regulatory effects.

- **DE Spearman Significance (DES):** This assesses the rank consistency of log-fold changes (LFC) specifically for genes identified as statistically significant via the Wilcoxon rank-sum test ($p < 0.05$).

$$\mathrm{DES} = \frac{1}{|\mathcal{P}|} \sum_{p \in \mathcal{P}} \mathrm{Spearman}\left(\log_2 \frac{\hat{\mathbf{y}}_{p,\mathrm{sig}}}{\bar{\mathbf{y}}_{\mathrm{ctrl,sig}}}, \log_2 \frac{\mathbf{y}_{p,\mathrm{sig}}}{\bar{\mathbf{y}}_{\mathrm{ctrl,sig}}}\right) \tag{14}$$

**Distributional Discriminative Power:**

- **Perturbation Discrimination Score (PDS):** To evaluate if the predicted state $\hat{\mathbf{y}}_p$ is uniquely identifiable, we compute

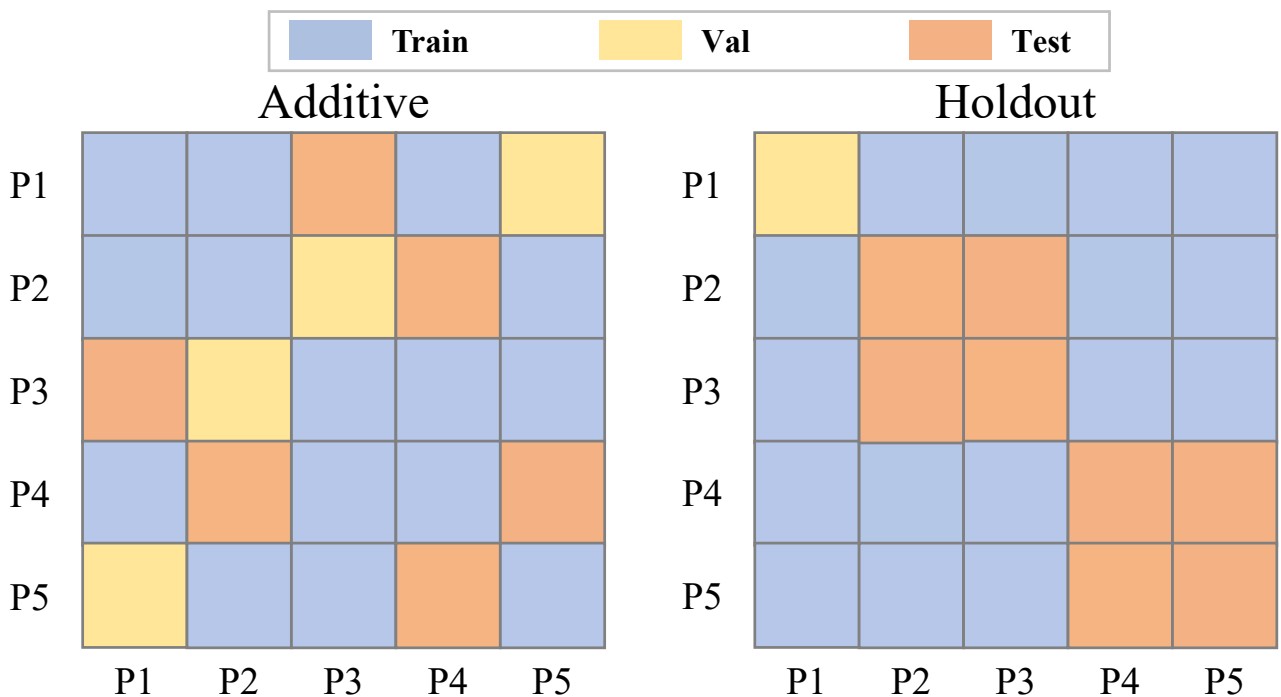

*Figure 8.* **Overview of experimental settings. Additive setting**: Evaluates the model's ability to predict unseen dual-perturbation combinations based on observed single perturbations. **Holdout setting**: Assesses performance on both unseen single perturbations and entirely novel dual-perturbation combinations (§A.4.2).

the rank of the true target $\mathbf{y}_p$ among all possible true states $\{\mathbf{y}_t\}_{t=1}^{N}$ based on $L_1$ distance.

$$\text{PDS} = \frac{1}{|\mathcal{P}|} \sum_{p \in \mathcal{P}} \left( 1 - \frac{\text{rank}_p - 1}{N - 1} \right) \tag{15}$$

- **Energy Distance (E-Dist):** To compare the predicted and observed cell-level distributions for each perturbation, we compute the squared energy distance in a 50-dimensional PCA space. Let $\mathbf{Z}_p = \{\mathbf{z}_{p,i}\}_{i=1}^{n_p}$ and $\hat{\mathbf{Z}}_p = \{\hat{\mathbf{z}}_{p,j}\}_{j=1}^{m_p}$ denote PCA-projected observed and predicted cells under perturbation $p$:

$$
\begin{aligned}
\text{E-Dist}_p = {} & \frac{2}{n_p m_p} \sum_{i=1}^{n_p} \sum_{j=1}^{m_p} \|\mathbf{z}_{p,i} - \hat{\mathbf{z}}_{p,j}\|_2 \\
& - \frac{1}{n_p^2} \sum_{i=1}^{n_p} \sum_{i'=1}^{n_p} \|\mathbf{z}_{p,i} - \mathbf{z}_{p,i'}\|_2 - \frac{1}{m_p^2} \sum_{j=1}^{m_p} \sum_{j'=1}^{m_p} \|\hat{\mathbf{z}}_{p,j} - \hat{\mathbf{z}}_{p,j'}\|_2.
\end{aligned}
\tag{16}
$$

- **Wasserstein Distance:** We estimate the Wasserstein-1 distance between $\mathbf{Z}_p$ and $\hat{\mathbf{Z}}_p$ using entropic Sinkhorn optimal transport. With uniform weights $\mathbf{a}$ and $\mathbf{b}$ over observed and predicted cells and cost matrix $C_{ij} = \|\mathbf{z}_{p,i} - \hat{\mathbf{z}}_{p,j}\|_2$, the metric is:

$$W_p = \min_{\mathbf{T} \in U(\mathbf{a},\mathbf{b})} \langle \mathbf{T}, \mathbf{C} \rangle + \varepsilon \sum_{i,j} T_{ij} \left( \log T_{ij} - 1 \right), \tag{17}$$

where $\varepsilon = 1.0$ in our evaluation.

- **Discriminative Cosine Similarity (Discr. Cos):** This metric measures whether the predicted perturbation moves cells in the same transcriptional direction as the ground truth. For each perturbation, we compare the mean response vector

*Table 18.* Cross-cell-line transfer from Replogle_K562 to Replogle_RPE1. Higher is better for $\rho\Delta$, $\rho\Delta^{\mathrm{D}}$, and Discr. Cos; lower is better for MSE, E-Dist, and Wasserstein(§A.5).

| Method | $\rho\Delta\uparrow$ | $\rho\Delta^{\mathrm{D}}\uparrow$ | MSE↓ | Discr. Cos↑ | E-Dist↓ | Wasserstein↓ |
|---|---|---|---|---|---|---|
| CellFlow (Klein et al., 2025) | 0.409 | 0.531 | 0.0377 | 0.756 | 6.51 | 13.7 |
| scBIG (Ours) | **0.420** | **0.555** | **0.0324** | **0.764** | **5.37** | **13.2** |

relative to the control mean $\bar{\mathbf{x}}_{\mathrm{ctrl}}$:

$$\mathrm{DiscrCos}_p = \frac{(\bar{\mathbf{x}}_p - \bar{\mathbf{x}}_{\mathrm{ctrl}})^\top (\hat{\bar{\mathbf{x}}}_p - \bar{\mathbf{x}}_{\mathrm{ctrl}})}{\|\bar{\mathbf{x}}_p - \bar{\mathbf{x}}_{\mathrm{ctrl}}\|_2 \|\hat{\bar{\mathbf{x}}}_p - \bar{\mathbf{x}}_{\mathrm{ctrl}}\|_2}. \tag{18}$$

For all three distribution-level metrics, we report the macro-average over perturbations.

## A.5. Cross-Cell-Line Transfer

To evaluate transfer beyond standard single-cell-line benchmarks, we train scBIG on all Replogle_K562 perturbation data and test on held-out perturbations in Replogle_RPE1. As shown in Table 18, scBIG consistently improves over CellFlow across point-wise, directional, and distribution-level metrics, indicating robust generalization under a more heterogeneous cell-line shift.

## A.6. Baselines

To evaluate the effectiveness of scBIG, we benchmarked it against 13 baseline models. Unless otherwise specified, the baseline models were configured with their default parameters to ensure a fair comparison.

### A.6.1. CONTROL

The *Control* baseline serves as a null hypothesis, assuming that the biological system remains in a steady state despite the applied perturbation. Formally, it predicts the post-perturbation profile $\hat{\mathbf{x}}$ by directly adopting the empirical mean of the unperturbed control cells $\bar{\mathbf{x}}_{\mathrm{ctrl}}$:

$$\hat{\mathbf{x}} = \bar{\mathbf{x}}_{\mathrm{ctrl}} \tag{19}$$

This baseline effectively quantifies the baseline variance and provides a lower bound for measuring the magnitude of perturbation-induced transcriptional shifts.

### A.6.2. LINEAR

Following the benchmark framework established in (Ahlmann-Eltze et al., 2025), the *Linear* method models the transcriptomic response as a linear combination of gene-level latent features. Let $\mathbf{G}$ be the gene embedding matrix derived from the top $K$ ($K = 512$) principal components of the training data $\mathbf{Y}^{\mathrm{train}}$. The model optimizes a weight matrix $\mathbf{W}$ to minimize the reconstruction error:

$$\mathbf{W}^* = \arg\min_{\mathbf{W}} \|\mathbf{Y}^{\mathrm{train}} - (\mathbf{G}\mathbf{W}^T + \mathbf{b})\|_2^2 \tag{20}$$

where $\mathbf{b}$ denotes the mean expression vector. The optimal $\mathbf{W}$ is solved via normal equations with Tikhonov regularization ($\lambda = 0.1$) to ensure numerical stability and generalization to unseen perturbations.

### A.6.3. LINEAR-SCGPT

The Linear-scGPT (Ahlmann-Eltze et al., 2025) baseline adopts an identical regression architecture to the Linear method but replaces the statistical PCA components with biologically-informed embeddings. Specifically, the matrix $\mathbf{G}$ is populated with high-dimensional gene representations extracted from the scGPT (Cui et al., 2024) foundation model . By leveraging zero-shot embeddings that capture complex gene-gene semantic relationships, this method assesses the extent to which pre-trained biological priors can enhance linear prediction accuracy.

### A.6.4. GEARS

GEARS (Roohani et al., 2024) integrates gene-gene interaction graphs with a multi-layer perceptron to predict multi-gene combinatorial perturbations. We adopted the standard GEARS architecture and its built-in gene association graphs. The

model was trained using the Adam optimizer with a learning rate of $10^{-3}$ for 20 epochs.

### A.6.5. CELLORACLE

CellOracle (Kamimoto et al., 2023) facilitates *in silico* gene perturbations by integrating Gene Regulatory Network (GRN) inference with vector field analysis. We utilized the official implementation from the CellOracle repository to construct cluster-specific networks. To accommodate the smaller cell count in our control group relative to the standard tutorial datasets, we adjusted the computational parameters by reducing the number of iterations ($n\_iters$) from 100,000 to 1,000, thereby improving computational efficiency while maintaining the integrity of the regulatory simulations.

### A.6.6. SAMSVAE

samsVAE (Wu et al., 2022) leverages a variational autoencoder architecture that decouples the latent space to isolate perturbation effects from background cellular states. We applied the default KL-divergence weights and network depths during training to ensure stability in learning low-dimensional representations.

### A.6.7. GRAPHVCI

GraphVCI (Bereket & Karaletsos, 2023) utilizes variational causal inference and graph neural networks to model the impact of perturbations on gene networks. We executed its official code library while maintaining the default hidden layer dimensions and regularization parameters to evaluate its recovery performance under structural constraints.

### A.6.8. SCGPT

scGPT (Cui et al., 2024) is a generative foundation model built on the Transformer architecture, pre-trained on millions of single cells to capture complex gene regulatory networks. For perturbation prediction, we followed the official repository's protocol for encoding perturbation conditions. The model was fine-tuned on task-specific datasets for 20 epochs using default hyperparameters, with the best-performing checkpoint selected based on validation results.

### A.6.9. SCFOUNDATION

scFoundation (Hao et al., 2024) is a large-scale model pre-trained on the human cell atlas, providing high-capacity gene expression embeddings. In our experiments, we utilized its pre-trained weights to extract gene embeddings and employed the GEARS (Roohani et al., 2024) framework for downstream training. The training process followed the default pipeline, ensuring that data normalization was strictly aligned with the pre-training phase.

### A.6.10. GENECOMPASS

GeneCompass (Yang et al., 2024) integrates cross-species transcriptomic data to enhance sensitivity to genetic perturbations through knowledge-augmented pre-training. We utilized its pre-trained weights from the human dataset to extract embeddings and performed training within the GEARS (Roohani et al., 2024) framework. The implementation maintained the recommended optimizer settings and batch sizes as specified in the official code.

### A.6.11. CELLFM

CellFM (Zeng et al., 2025) is a large-scale foundation model pre-trained on an extensive atlas of 100 million human single-cell transcriptomes, leveraging the Transformer architecture to learn comprehensive gene representations across diverse biological contexts. In our experimental framework, we utilized the pre-trained CellFM weights as a high-capacity feature extractor to derive gene-level embeddings. These biologically-informed embeddings were subsequently integrated into the GEARS (Roohani et al., 2024) model for downstream perturbation prediction. The integrated model was trained for 20 epochs using the default optimization settings to effectively align the foundation model's prior knowledge with the specific regulatory dynamics of the target dataset.

### A.6.12. STATE

STATE (Adduri et al., 2025) models cellular responses as transitions within a latent manifold. In our benchmarking, we trained the STATE model from scratch using the official implementation to ensure a fair and reproducible experimental setup. We opted against using the pre-trained weights because the publicly available transition models are specifically

---

**Algorithm 1** Gene-Relation Clustering (GRC)

---

1: **Output:** Balanced hard partition $\mathcal{C} = \{c_1, \ldots, c_K\}$.
2:
3: Initialize centroids $\{\mu_k\}_{k=1}^K$ via $k$-means++ on $\mathbf{E}$.
4: Assign initial clusters $\mathcal{C}$ using $D^{\text{sem}}$.
5:
6: **for** $t = 1$ **to** $T$ **do**
7:     {// Update Cost Matrix with PPI prior}
8:     **for** $i = 1$ **to** $G$, $k = 1$ **to** $K$ **do**
9:         $d_{ik}^{\text{sem}} = 1 - \text{cosine}(E_i, \mu_k)$
10:         $d_{ik}^{\text{PPI}} = 1 - \frac{1}{|c_k|} \sum_{j \in c_k} A_{ij}^{\text{PPI}}$
11:         $C_{ik} = d_{ik}^{\text{sem}} + d_{ik}^{\text{PPI}}$
12:     **end for**
13:
14:     {// Optimal Transport via Sinkhorn}
15:     Set $a = \frac{1}{G}\mathbf{1}_G, b = \frac{1}{K}\mathbf{1}_K$.
16:     $\mathbf{\Pi}^* = \text{Sinkhorn}(\mathbf{C}, a, b, \epsilon)$
17:
18:     {// Balanced Rounding & Update}
19:     $\mathcal{C} \leftarrow \text{CapacityConstrainedRounding}(\mathbf{\Pi}^*)$
20:     $\mu_k \leftarrow \text{mean}(\{E_i \mid i \in c_k\})$ for each $k$.
21: **end for**
22:
23: **Return** Final partition $\mathcal{C}$.

---

formulated for single-agent "drug-dosage" scenarios and do not natively support the gene perturbations or combinatorial settings evaluated in this work.

### A.6.13. CELLFLOW

CellFlow (Klein et al., 2025) models perturbation responses as continuous transformations between control and perturbed states using a flow-matching objective to learn a velocity field in a low-dimensional latent space. Following the official reference implementation, we first projected the gene expression data into a 50-dimensional PCA space to serve as the latent representation. During inference, the model integrates the learned flow to predict the perturbed state, which is subsequently mapped back to the full gene space via the PCA decoder for evaluation.

### A.7. Pathway Analysis of Gene Clusters

To provide a more granular view of the biological identity of each gene module, we performed an extensive pathway enrichment analysis at the sub-pathway level for both the *Norman* and *Repl* datasets. Complementing the categorical summary in the main text, Figures 9 and 10 illustrate the $-\log_{10}(p\text{-value})$ of specific Reactome sub-pathways across the 32 gene clusters identified by GRC.

In the *Norman* dataset (see Figure 9), the sub-pathway distribution reveals highly specialized functional modules. For instance, Cluster 9 (C9) exhibits extreme enrichment in core cell cycle regulators such as *E2F targets*, *G2/M Checkpoint*, and *Mitotic Spindle Checkpoint*, while Cluster 3 (C3) is distinctly focused on *Eukaryotic Translation Termination*. Notably, Cluster 15 (C15) and Cluster 10 (C10) capture diverse RNA-related processes including *mRNA Splicing* and *Polyadenylation*, respectively.

In the *Repl* dataset (see Figure 10), where the genomic coverage is broader, we observe even more robust enrichment signatures. Cluster 15 (C15) shows a dominant association with fundamental translation machinery, including *Peptide chain elongation* and *Eukaryotic Translation Elongation*. Meanwhile, Cluster 25 (C25) and Cluster 28 (C28) effectively decouple different regulatory nodes of the cell cycle and splicing pathways.

These fine-grained heatmaps demonstrate that GRC does not merely group genes based on simple co-expression; instead, it

---

**Algorithm 2** Stage I: Self-supervised Pre-training (Latent Manifold Pre-training)

---

1: **Input:** Expression matrix $\mathbf{X} \in \mathbb{R}^{N \times G}$; gene clusters $\mathcal{C} = \{c_k\}_{k=1}^K$ from GRC; gene embeddings $\mathbf{E} \in \mathbb{R}^{G \times d}$; inducing points $\mathbf{I} \in \mathbb{R}^{M \times D}$.

2: **Output:** Pre-trained encoder $E_\phi$ (GCAE) and decoder $D_\psi$.

3: Initialize parameters $\phi, \psi$ (and $\mathbf{I}$ if learnable).

4: **repeat**

5:     Sample a batch $\{\mathbf{x}_i\}_{i=1}^B$ from $\mathbf{X}$.

6:     **for** $i = 1$ **to** $B$ **do**

7:         **(1) Reorder into module-structured input.**

8:         $\mathbf{x}_i^{\mathrm{mod}} \leftarrow \textsc{ReorderByClusters}(\mathbf{x}_i, \mathcal{C}) \in \mathbb{R}^{K \times L}, \; L = G/K.$

9:         **(2) Dual-stream fusion (expression + semantic priors).**

10:         $\mathbf{h}_k^{\mathrm{exp}} \leftarrow \mathrm{Proj}_{\mathrm{exp}}(\mathbf{x}_{i,k}^{\mathrm{mod}}) \in \mathbb{R}^D, \; \text{for } k = 1..K.$

11:         $\bar{\mathbf{e}}_k \leftarrow \frac{1}{|c_k|} \sum_{g \in c_k} \mathbf{E}_g, \; \mathbf{h}_k^{\mathrm{sem}} \leftarrow \mathrm{Proj}_{\mathrm{sem}}(\bar{\mathbf{e}}_k) \in \mathbb{R}^D.$

12:         $\mathbf{H}_i^{(0)} \leftarrow [\mathbf{h}_1^{\mathrm{exp}} + \mathbf{h}_1^{\mathrm{sem}}; \dots; \mathbf{h}_K^{\mathrm{exp}} + \mathbf{h}_K^{\mathrm{sem}}] + \mathrm{PosEnc}(\mathcal{C}) \in \mathbb{R}^{K \times D}.$

13:         **(3) Perceiver bottleneck: Compress–Process–Broadcast.**

14:         $\mathbf{I}' \leftarrow \mathrm{LN}(\mathbf{I} + \mathrm{CrossAttn}(\mathbf{I}, \mathbf{H}_i^{(0)})).$

15:         $\mathbf{I}'' \leftarrow \mathrm{LN}(\mathbf{I}' + \mathrm{SelfAttn}(\mathbf{I}')).$

16:         $\mathbf{H}_i^{(1)} \leftarrow \mathrm{LN}(\mathbf{H}_i^{(0)} + \mathrm{CrossAttn}(\mathbf{H}_i^{(0)}, \mathbf{I}'')).$

17:         **(4) Cell embedding + reconstruction.**

18:         $\mathbf{z}_i \leftarrow \mathrm{Linear}\left(\frac{1}{K} \sum_{k=1}^K \mathbf{H}_{i,k}^{(1)}\right) \in \mathbb{R}^D.$

19:         $\hat{\mathbf{x}}_i \leftarrow D_\psi(\mathbf{z}_i) \in \mathbb{R}^G.$

20:     **end for**

21:     $\mathcal{L}_{\mathrm{recon}} \leftarrow \frac{1}{B} \sum_{i=1}^B \|\hat{\mathbf{x}}_i - \mathbf{x}_i\|_2^2.$

22:     Update $\phi, \psi$ (and $\mathbf{I}$ if applicable) by Adam to minimize $\mathcal{L}_{\mathrm{recon}}$.

23: **until** convergence

---

successfully partitions them into discrete, biologically interpretable units that correspond to specific biochemical cascades. By mapping genes to these functional modules, GRC enables the model to more effectively capture and learn the intricate interactions between distinct biological modules.

## A.8. Limitations and Future Work

**Zero-shot Generalization and Evaluation Scope.** A key goal of this work is zero-shot prediction under unseen perturbations. We evaluate this capability on two representative Perturb-seq benchmarks, Norman and RPE1, and explicitly focus on held-out perturbations to probe generalization. While these results provide evidence of transfer beyond seen interventions, they do not fully cover the diversity of real-world perturbation regimes and experimental settings. Extending evaluation to broader multi-context resources—spanning additional cell types, protocols, and intervention classes—will be important for establishing robustness at scale.

**Gene Programs and Module Induction.** Our approach emphasizes program-level organization by inducing gene modules via Gene-Relation Clustering (GRC) and using them as a structured scaffold. This design improves interpretability and supports efficient module-aware attention, but the induced partition remains a simplified and largely static approximation of gene-program structure. Module granularity (*e.g.*, the choice of $K$) and the dependence on external signals (pretrained gene embeddings and PPI priors) may not fully capture context-specific programs that shift across cell states or perturbation regimes. An important direction for future work is to develop context-adaptive or learnable module induction mechanisms, allowing program structure to adjust to condition-specific data.

**Structural Priors and Foundation Representations.** scBIG incorporates multiple sources of prior information, including pretrained gene embeddings, PPI-derived structure, curated pathway memberships, and ESM2-based perturbation embeddings. These components provide strong biological grounding, but they also make performance sensitive to the quality, coverage, and context-mismatch of the underlying resources. Static priors can be incomplete, and representation errors can propagate into module induction and the biological-consistency objectives. We view scBIG as a flexible framework where

---

**Algorithm 3** Stage II: End-to-End Conditional Flow Matching with Structure-Aware Alignment

---

1: **Input:** Pre-trained encoder/decoder $(E_\phi, D_\psi)$; vector field $v_\theta$; training pairs $(\mathbf{x}_0, \mathbf{x}_1, p)$; perturbation embedding $\mathbf{c}(p)$ (ESM2); clusters $\mathcal{C}$; pathway membership matrix $\mathbf{S} \in \{0, 1\}^{P \times G}$; weights $(\lambda_{\text{corr}}, \lambda_{\text{pathway}})$; OT regularization $\epsilon$.

2: **Output:** Fine-tuned parameters $\{\phi, \psi, \theta\}$ (scBIG).

3: **Precompute / cache target statistics for efficiency:**

4: For each perturbation condition $p$: compute and cache $\mathbf{R}_{gt}^*(p)$ and $\mathbf{y}_{gt}^*(p)$.

5:    $\mathbf{R}_{gt}^*(p) \leftarrow \text{PEARSONCORR}(\text{CLUSTERAGG}(\mathbf{x}_1, \mathcal{C}))$   (cluster-level correlation target).

6:    $\mathbf{y}_{gt}^*(p) \leftarrow \mathbf{S}\,\mathbf{x}_1$   (pathway activation target; optionally averaged over cells with condition $p$).

7: **repeat**

8:    Sample a batch $\{(\mathbf{x}_0, \mathbf{x}_1, p)_i\}_{i=1}^B$ and sample $t_i \sim \mathcal{U}(0, 1)$.

9:    **for** $i = 1$ **to** $B$ **do**

10:      **(1) Encode control/perturbed cells.**

11:      $\mathbf{z}_0 \leftarrow E_\phi(\mathbf{x}_0), \quad \mathbf{z}_1 \leftarrow E_\phi(\mathbf{x}_1)$.

12:      **(2) Flow matching regression on linear path.**

13:      $\mathbf{z}_t \leftarrow (1 - t)\mathbf{z}_0 + t\mathbf{z}_1$.

14:      $\mathbf{u} \leftarrow (\mathbf{z}_1 - \mathbf{z}_0), \quad \hat{\mathbf{u}} \leftarrow v_\theta(\mathbf{z}_t, t, \mathbf{c}(p))$.

15:      $\mathcal{L}_{\text{flow}}^{(i)} \leftarrow \|\hat{\mathbf{u}} - \mathbf{u}\|_2^2$.

16:      **(3) Decode predicted endpoint via ODE solve.**

17:      $\hat{\mathbf{z}}_1 \leftarrow \text{ODESOLVE}(\mathbf{z}_0, v_\theta, \mathbf{c}(p))$   (integrate to $t{=}1$).

18:      $\hat{\mathbf{x}}_1 \leftarrow D_\psi(\hat{\mathbf{z}}_1)$.

19:      **(4) Cluster Correlation Alignment loss $\mathcal{L}_{\text{corr}}$.**

20:      $\bar{\hat{\mathbf{x}}}_1 \leftarrow \text{CLUSTERAGG}(\hat{\mathbf{x}}_1, \mathcal{C}) \in \mathbb{R}^K$   (*e.g.*, mean per cluster).

21:      $\mathbf{R}(\bar{\hat{\mathbf{x}}}_1) \leftarrow \text{PEARSONCORR}(\bar{\hat{\mathbf{x}}}_1) \in \mathbb{R}^{K \times K}$.

22:      $\mathcal{L}_{\text{corr}}^{(i)} \leftarrow \|\mathbf{R}(\bar{\hat{\mathbf{x}}}_1) - \mathbf{R}_{gt}^*(p)\|_F$.

23:      **(5) Pathway-informed OT loss $\mathcal{L}_{\text{pathway}}$.**

24:      $\hat{\mathbf{y}} \leftarrow \mathbf{S}\hat{\mathbf{x}}_1 \in \mathbb{R}^P, \quad \mathbf{y}^* \leftarrow \mathbf{y}_{gt}^*(p)$.

25:      $\mathcal{L}_{\text{pathway}}^{(i)} \leftarrow \mathcal{W}_\epsilon(\hat{\mathbf{y}}, \mathbf{y}^*)$   (Sinkhorn OT distance).

26:      **end for**

27:    $\mathcal{L}_{\text{flow}} \leftarrow \frac{1}{B} \sum_{i=1}^B \mathcal{L}_{\text{flow}}^{(i)}, \mathcal{L}_{\text{corr}} \leftarrow \frac{1}{B} \sum_{i=1}^B \mathcal{L}_{\text{corr}}^{(i)}, \mathcal{L}_{\text{pathway}} \leftarrow \frac{1}{B} \sum_{i=1}^B \mathcal{L}_{\text{pathway}}^{(i)}$.

28:    $\mathcal{L}_{\text{total}} \leftarrow \mathcal{L}_{\text{flow}} + \lambda_{\text{corr}}\mathcal{L}_{\text{corr}} + \lambda_{\text{pathway}}\mathcal{L}_{\text{pathway}}$.

29:    Update $\{\phi, \psi, \theta\}$ by Adam to minimize $\mathcal{L}_{\text{total}}$.

30: **until** convergence

---

such priors are modular components; future iterations could incorporate confidence-aware weighting, context-specific graph construction, or data-driven refinement of priors to improve robustness.

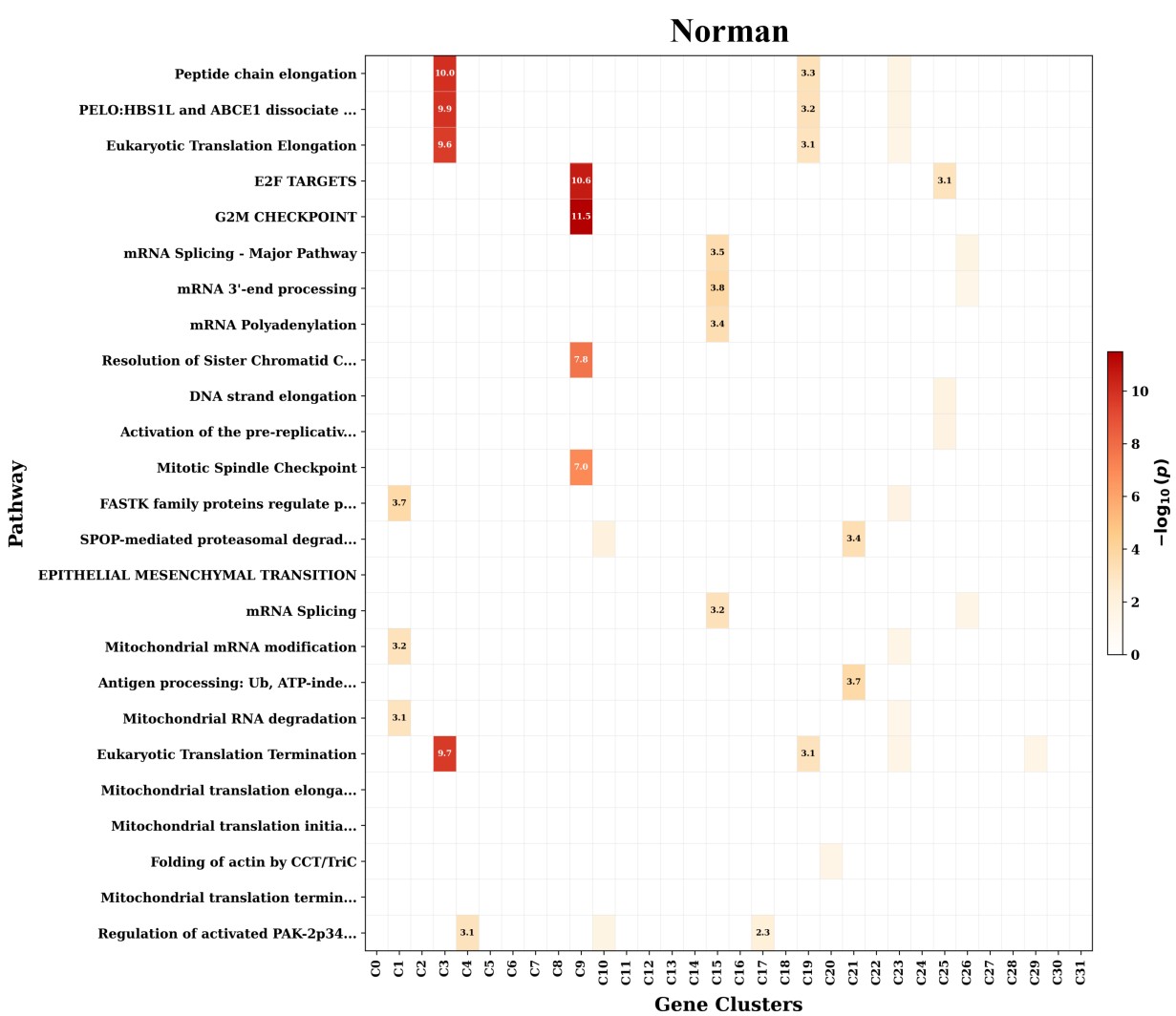

*Figure 9.* Pathway enrichment of gene clusters on the Norman dataset (§A.7).

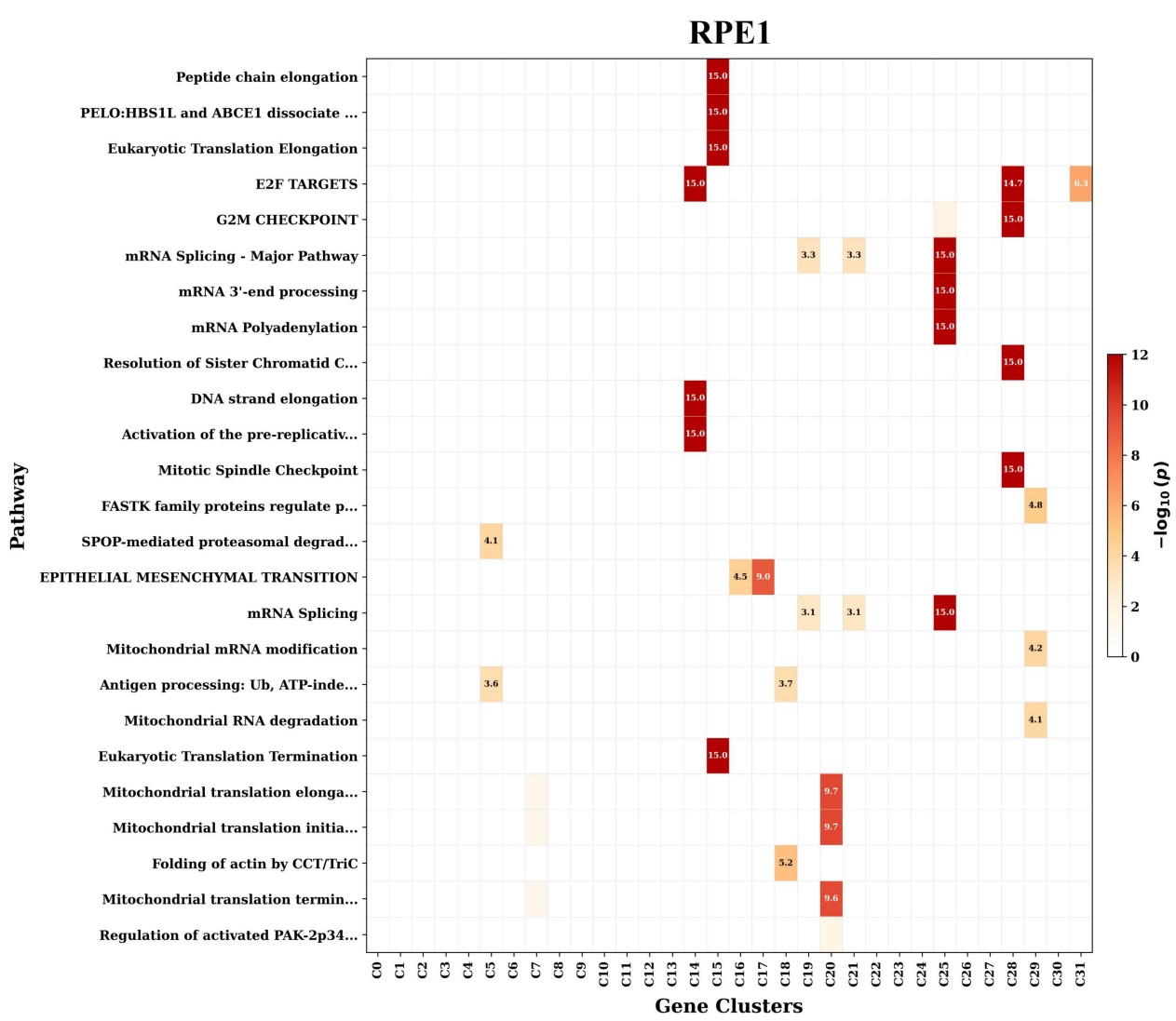

*Figure 10.* Pathway enrichment of gene clusters on the RPE1 dataset (§A.7).

