# OpenReview forum: "Beyond Independent Genes: Learning Module-Inductive Representations for Single-Cell Gene Perturbation Prediction"
_ICML.cc/2026/Conference — ICML 2026 regular_

### Official Review · Reviewer_j2pN · 2026-03-12

**Soundness:** 2
**Presentation:** 3
**Significance:** 3
**Originality:** 3
**Overall Recommendation:** 4
**Confidence:** 4

**Summary:**

This paper proposes scBIG, a module-inductive generative framework for predicting transcriptional responses to genetic perturbations. The key idea is to move beyond gene-wise modeling by explicitly inducing and reasoning over coordinated gene programs. Experiments on the Norman and RPE1 benchmarks demonstrate consistent improvements over several baselines across multiple metrics.  The paper presents a well-motivated and technically coherent approach to an important problem in computational biology. However, the evaluation scope is notably narrow compared to the baselines it benchmarks against. Additionally, the reliance on mean-based evaluation metrics undermines the justification for using a generative model. In its current form, the paper makes a solid but somewhat incremental contribution within a restricted experimental scope.

**Compliance With Llm Reviewing Policy:**

Affirmed.

**Final Justification:**

The authors provided additional experiments in the rebuttal phase. Several of my concerns have been addressed by the new results. Therefore, I decide to increase my score to 4.

**Key Questions For Authors:**

1. Why is CPA (Compositional Perturbation Autoencoder) not included as a baseline?
2. What is the relationship between GRC module assignments and the attention patterns in GCAE?

**Limitations:**

Yes

**Strengths And Weaknesses:**

**Strength:**

1. The core idea of modeling perturbation responses at the gene module level rather than treating genes independently is biologically well-grounded.
2. scBIG consistently achieves state-of-the-art results across all evaluated metrics on both the Norman and RPE1 benchmarks, outperforming 13 baselines spanning four different paradigms.
3. The paper provides a comprehensive set of ablation experiments that systematically isolate the contribution of each component, including clustering strategies, encoding architectures, biological regularization terms, loss weight sensitivity, gene embedding choices, and structural hyper-parameters.



**Weakness:**

[major] Limited generalizability and narrow task scope compared to baselines.

Although scBIG achieves strong performance on the genetic perturbation prediction task compared to baseline methods such as CellFlow, STATE, and scGPT, it is worth noting that these baselines are substantially more general-purpose frameworks. CellFlow and STATE, for instance, have been evaluated on a diverse range of perturbation modalities, and have demonstrated the ability to generalize to novel cell lines. In contrast, scBIG is evaluated exclusively on single-cell-line settings (K562 for Norman, RPE1 for Replogle), The narrow evaluation scope, restricted to genetic perturbations in single-cell-line settings, makes scBIG appear somewhat limited in comparison to the more versatile baselines it claims to outperform.


[major] Mismatch between generative modeling motivation and evaluation metrics.

A central motivation of scBIG is the use of a flow-based generative model to capture the distributional heterogeneity of single-cell perturbation responses, as opposed to deterministic prediction approaches. However, all reported evaluation metrics are computed on mean expression profiles (pseudobulk-level), which only assess the accuracy of average predicted effects and entirely ignore the distributional quality of generated cell populations. This creates a fundamental disconnect between what the model is designed to do and what the evaluation actually measures. I would strongly recommend the authors report distributional metrics such as energy distance, Wasserstein distance.

[minor] Circular reasoning in the validation of data-driven gene modules.

A key claim of scBIG is that data-driven gene module induction via GRC is superior to relying on static biological priors such as Gene Ontology or predefined pathways. The authors argue that such priors are "static, incomplete, and context-agnostic" (Section 1), and position GRC as a more flexible alternative that can capture dynamic, context-specific gene program organization. However, the primary evidence used to validate the quality of GRC-induced modules is pathway enrichment analysis against the Reactome database (Section 3.4, Figures 3, 8, 9), which is itself a canonical, static biological prior of exactly the kind the paper criticizes. In other words, the authors first argue that static pathway knowledge is insufficient for modeling perturbation responses, but then use agreement with that same static pathway knowledge as the gold standard for evaluating whether their learned modules are biologically meaningful. A more convincing validation would involve demonstrating that GRC modules capture regulatory structures that are not well-represented by existing pathway databases, like novel co-regulation patterns specific to particular perturbation conditions.

---

> ### Author Rebuttal · Authors · 2026-03-30
>
> Thank you for the insightful and encouraging feedback. We add new experiments and clarifications below and **respectfully ask whether they would merit an updated score.**
>
> ---
>
> ***Q1: Generalizability beyond single-cell-line genetics?***
>
> **A1:** Thank you for your suggestions. **scBIG can indeed generalize across drug perturbations and cell lines, and it indeed outperforms CPA[1] (you proposed)**. We add two tests: unseen-drug prediction on ComboSciPlex[1] (Table1) and cross-cell-line transfer from Replogle_K562[2] to held-out Replogle_RPE1 perturbations (Table2). For fair drug comparison, all methods use ChemBERTa[3] as the drug encoder.
>
> >Table 1: Drug perturbation.
>
> |Method|ρΔ|ρΔ^D|MSE|Disc.Cos|E-Dis|Wass.|
> |---|---:|---:|---:|---:|---:|---:|
> |scGPT|0.8172|0.9280|0.0597|0.8030|14.12|17.46|
> |CPA|0.8387|0.9003|0.0652|0.8388|10.17|18.15|
> |STATE|0.6233|0.7418|0.0904|0.6145|13.46|17.91|
> |CellFlow|0.8276|0.9189|0.0461|0.8178|4.83|17.53|
> |scBIG|0.8671|0.9321|0.0451|0.8612|4.74|16.76|
>
> scBIG is best on both mean-level and distribution-level metrics, including against CPA, a drug-focused baseline[1].
>
> >Table 2: Cross-cell-line transfer.
>
> |Method|ρΔ|ρΔ^D|MSE|Disc.Cos|E-Dis|Wass.|
> |---|---:|---:|---:|---:|---:|---:|
> |CellFlow|0.4086|0.5310|0.0377|0.7555|6.51|13.66|
> |scBIG|0.4202|0.5546|0.0324|0.7642|5.37|13.15|
>
> scBIG outperforms CellFlow across cell lines. **Using STATE official code, we obtain ρΔ=0.013 on this task; as also reported by others, reproduction differs from the paper report** (https://github.com/ArcInstitute/state/issues/254). **We will add these results in experiments.**
>
> ***Q2: Generative model but mean-only evaluation?***
>
> **A2:** Thank you for the insightful suggestions. **scBIG also performs strongly on distribution metrics.** We evaluate distributional metrics on Table 1-2 and the Norman additive split (Table3): discriminator cosine similarity, energy distance, and Wasserstein distance.
>
> >Table 3: Distribution-level evaluation.
>
> |Method|Disc.Cos|E-Dis|Wass.|
> |---|---:|---:|---:|
> |Control|-0.3009|13.5125|16.5615|
> |Linear|0.3967|11.1376|15.3741|
> |GEARS|0.5871|1.2377|11.0829|
> |scFoundation|0.6218|1.3710|11.2452|
> |GeneCompass|0.5623|1.3059|11.1429|
> |CellFlow|0.6397|1.1979|10.7579|
> |scBIG|0.7264|0.8383|9.7842|
>
> scBIG is best on all three metrics, outperforming the strongest prior generative baseline CellFlow. **We will revise the paper to present these.**
>
> ***Q3: Is Reactome-based validation circular? Novel Coordination?***
>
> **A3:** No, we use static priors as a partial external audit, not as the main predictive scaffold. **As stated in Sec. 2.1, curated priors are high-confidence but low-coverage**: insufficient for modeling perturbation responses, yet still useful for checking whether learned modules recover known biology.
>
> Beyond this audit, **scBIG also identifies perturbation-specific cross-module coordination not directly specified by the priors (Table4)**.
>
> >Table 4: Candidate coordination beyond audited priors. **Module identities are at https://anonymous.4open.science/r/scBIG-4A15/README_j2pN.md**.
>
> |Pair|Perturb.|Coact|Boot|
> |---|---|---:|---:|
> |M2-M30|CEBPE+ZC3HAV1|0.57|1.00|
> |M2-M8|OSR2+UBASH3B|0.52|0.98|
> |M2-M7|OSR2+UBASH3B|0.42|0.97|
>
> `Coact` is positive coactivation across perturbations and `Boot` its bootstrap positive rate.
> **For `CEBPE+ZC3HAV1`, `M2` is chromatin/proteostasis-like and `M30` cytoskeleton/growth-like, suggesting coupled differentiation/stress rewiring and a downstream cytoskeletal-growth response, consistent with prior chromatin-cytoskeleton coupling work[4]. We agree this is stronger validation and will emphasize it in Sec. 3.4.**
>
> ***Q4: Why not include `CPA`?***
>
> **A4:** `CPA` can be competitive on smoother drug perturbations (Table1), but it is much weaker on combinatorial genetic perturbations (Table5) **because it learns perturbation-to-expression mappings without explicitly modeling module coordination.**
>
> >Table 5: `CPA` vs. `scBIG`.
>
> |Method|ρΔ|ρΔ^D|ACCΔ|ACCΔ^D|DES|L2|Disc.Cos|E-Dist|Wass.|
> |---|---:|---:|---:|---:|---:|---:|---:|---:|---:|
> |CPA|0.4159|0.6320|0.5388|0.8016|0.4827|9.30|0.2129|4.3680|13.1650|
> |scBIG|0.8496|0.8230|0.9197|0.9906|0.8593|3.92|0.7264|0.8383|9.7842|
>
> scBIG outperforms `CPA` on both mean-level and distribution-level metrics, and **we will add this comparison in the Sec. 3.1.**
>
> ***Q5: GRC modules vs. GCAE attention?***
>
> **A5:** As described in Sec. 2.1-2.2, `GRC` defines the biologically informed module scaffold, while `GCAE` models context-dependent interactions on top of it. Thus, `GCAE` attention is not independent of `GRC`: `GRC` defines the functional units, and `GCAE` models how they coordinate under perturbation. For better clarity, we will make this more explicit in the revision.
>
> [1] Lotfollahi et al., *Mol Syst Biol*, 2023.
>
> [2] Replogle et al. Cell 2022.
>
> [3] Chithrananda et al., arXiv, 2020.
>
> [4] Alam et al., *Sci Rep*, 2016.
>
> ---
>
> **We hope these additions address your concerns and would support an updated score.**

---

> > ### Author Rebuttal · Reviewer_j2pN · 2026-04-03
> >
> > Thanks to the authors for conducting additional experiments. Most of my concerns have been addressed by the new results, and I appreciate the effort put into strengthening the work. Accordingly, I have increased my score to 4.

---

> > > ### Author Response · Authors · 2026-04-04
> > >
> > > We sincerely thank the reviewer for the encouraging feedback. We are glad that our responses have addressed most of your concerns.
> > >
> > > Your suggestions on broader generalization, distribution-level evaluation, and module validation have been very helpful in strengthening both the evaluation and the presentation of the paper. If there are any remaining minor concerns or points that would benefit from further clarification, we would be very happy to address them and incorporate the corresponding improvements in the revision.
> > >
> > > Thank you again for your valuable time and for the positive shift in your assessment.

---

### Official Review · Reviewer_Vjfe · 2026-03-12

**Soundness:** 4
**Presentation:** 3
**Significance:** 4
**Originality:** 4
**Overall Recommendation:** 5
**Confidence:** 4

**Summary:**

This paper proposes scBIG, a  generative framework for predicting transcriptional responses to gene perturbations at single cell resolution. The goal is to model structured, program level perturbation responses in a generalizable and biologically coherent way. The work presents the concept of inducing gene modules directly from data and then building a hierarchical generative model that preserves coordinated behavior across gene modules. scBIG introduces three major components: i) Gene-Relation Clustering (GRC) to induce functional modules using semantic gene embeddings and PPI information via optimal transport. ii) Gene-Cluster-Aware Encoder (GCAE) to model interactions across modules using a Perceiver-style latent bottleneck. And iii) structure-Aware Alignment with two biological consistency losses (cluster correlation alignment and pathway-informed optimal transport).

**Compliance With Llm Reviewing Policy:**

Affirmed.

**Final Justification:**

I maintain my accept recommendation. The authors have thoroughly addressed the main technical and presentation concerns by clarifying the role and semantics of inducing points, motivating the balanced clustering as an architectural regularization rather than a biological claim, and providing additional ablation studies, confidence intervals, and statistical tests that strengthen the empirical conclusions. The added analyses on scalability, compute efficiency, and cross‑modality as well as cross‑cell‑line generalization further reinforce the robustness and practical relevance of the approach. Overall, scBIG presents a technically sound, biologically well‑motivated, and original framework for single‑cell perturbation modeling, with comprehensive experimental validation and clear potential impact, meeting the acceptance bar for ICML.

**Key Questions For Authors:**

-   How sensitive are results to the balanced constraint in GRC? Can you report performance when allowing variable cluster sizes, and provide biological coherence metrics (e.g., intra‑cluster PPI density/enrichment) vs. performance trade‑offs?
-  Beyond the appendix sweep, can you provide a principled heuristic or validation‑time criterion to select K (number of modules) and M (inducing points) across datasets, and quantify the variance in outcomes across seeds/choices?
-  In Eq. (3), please clarify the initialization and semantics of the inducing points I (distribution, scale), whether they are learned per model or per batch, and why mean‑pooling in Eq. (4) is preferred over attention/Set transformers for generating the cell‑level latent Z.
- Tables 1–3 show strong averages. Could you add confidence intervals / standard errors (not only std. dev. across seeds in Appx) and statistical tests vs. top baselines (e.g., paired tests across perturbations) to substantiate claims like “average +6.7% improvement”?
-    Please summarize wall‑clock, GPU memory, and throughput for training/inference under representative settings in the main text, and discuss scaling for larger gene sets (e.g., >2k HVGs) or multi‑context datasets.
-   Have you tested scBIG on additional perturbation modalities (e.g., small‑molecule, dose–response) or cell types to assess cross‑context generalization?

**Limitations:**

yes

**Strengths And Weaknesses:**

Strengths:
-  The method addresses an important  challenge in perturbation modeling and presents strong biological motivation
-  The proposed method is novel with a well-designed architecture
-   The combination of Cluster Correlation Alignment, and Pathway Informed Optimal Transport provides structure-aware biological objectives
-   Experiments and evaluations are comprehensive and solid

Weaknesses:
-  Eq. (3) uses inducing points I as queries in cross‑attention; however, the initialization/semantics of I and rationale for specific M are mostly deferred to appendices and could be clearer in the main text. The benefit of Eq. (4)’s cluster‑pooled cell embedding vs. alternatives (e.g., attention pooling) could be better justified.
-  The strictly balanced clustering (|cₖ|=G/K) risks noisy assignments when biology is imbalanced. The paper partly addresses this via ablations (soft vs. balanced) but the biological impact of forced balancing and sensitivity to K need more front‑matter emphasis (some appears in Appx A.1.1).
-  While Table 4 (a–e) provides ablations, the incremental effect sizes of GRC vs. K‑means/“unordered baseline” are sometimes modest (and without CIs), making it hard to quantify how much each component contributes under statistical uncertainty.
-  Compute and scalability discussion is not explained in the main paper. Efficiency numbers and overhead analyses are mostly in the appendix; a more practical compute profile (wall‑time, memory) for typical dataset scales and for varying K, M would help practitioners.
-   The narrative highlights GeneCompass for clustering, yet an ablation reports ESM2 as superior for perturbation conditioning; the paper notes a hypothesis, but this warrants clearer exposition and a consolidated takeaway to avoid confusion.

---

> ### Author Rebuttal · Authors · 2026-03-29
>
> We sincerely appreciate your insightful and professional comments. In addition to your acknowledgement of our paper’s main contributions, we have carefully considered your constructive feedback and questions with more ablation experiments.
>
> ---
> ***Q1: Inducing points and pooling?***
>
> **A1:** `I` are learned model parameters shared across samples, not batch-specific variables. They serve as latent slots that summarize gene chunks; what varies per sample is the attended context, not `I`. Because pooling is applied **after** this bottlenecked interaction, extra set-level machinery is unnecessary. The ablation in **[Table A1(Link)](https://anonymous.4open.science/r/scBIG-4A15/README_Vjfe.md)** confirms average pooling is the strongest readout under the same backbone. We will make this clearer in the revision.
>
> ***Q2: Balanced vs. soft clustering?***
>
> **A2:** Soft assignment is usable, but hard-balanced `GRC` is the better operating point: balancing prevents a few oversized clusters from absorbing heterogeneous genes, which preserves coherence without sacrificing prediction.
>
> >Table 1: Balanced vs. soft clustering.
>
> |Strategy|ρΔ|ρΔ^D|PPI dens.|Hallmark|
> |---|---:|---:|---:|---:|
> |Random(hard)|0.820|0.773|0.018|0.156|
> |K-means(soft)|0.825|0.777|0.083|0.188|
> |K-means(hard)|0.836|0.819|0.081|0.375|
> |GRC(soft)|0.832|0.828|0.118|0.281|
> |GRC(hard)|0.850|0.823|0.224|0.688|
>
> `GRC` stays competitive in both forms, but only the balanced version preserves clearly interpretable modules.
>
> ***Q3: How large and how certain is the `GRC` gain?***
>
> **A3:** We now add `95%` CIs. The right interpretation is not a step-change claim, but a stable ordering: unordered `<` generic grouping `<` biologically informed grouping, with `GRC+GCAE` best overall.
>
> >Table 2: Clustering/encoder comparison with `95%` CIs.
>
> |Clustering|Encoder|ρΔ|95% CI|
> |---|---|---:|---|
> |No clustering|GCAE|0.808|[0.754, 0.882]|
> |K-means|Normal|0.824|[0.769, 0.886]|
> |K-means|GCAE|0.836|[0.793, 0.894]|
> |GRC|Normal|0.835|[0.788, 0.895]|
> |GRC|GCAE|0.850|[0.799, 0.900]|
>
> ***Q4: How should `K` and `M` be selected?***
>
> **A4:** We search `K` on a small coarse grid, scale `M` with `K` (roughly `M ~= K/4`), and choose the **smallest** pair within a small validation margin of the best result. This avoids over-fragmentation. The new seed statistics in **[Table A2(Link)](https://anonymous.4open.science/r/scBIG-4A15/README_Vjfe.md)** show that the moderate regime remains most reliable, with `32/8` the strongest representative choice.
>
> ***Q5: How does the model scale?***
>
> **A5:** Wall-clock and memory are summarized in **[Table A3(Link)](https://anonymous.4open.science/r/scBIG-4A15/README_Vjfe.md)**. On a single `24 GB` 4090 GPU, `scBIG` remains practical from `2k` to `10k` HVGs by reducing batch size and increasing `K` and `M` together only when needed. Thus `32/8` is the default, while larger `K/M` is a capacity upgrade rather than a requirement.
>
> ***Q6: Why GeneCompass for clustering but ESM2 for perturbation encoding?***
>
> **A6:** We appreciate this suggestion and make it clearer in the revision. GeneCompass [1], pre-trained on massive single-cell transcriptomes, captures rich gene-context dependencies ideal for module induction. Conversely, ESM2 [2], pre-trained on evolutionary protein sequences, encodes fundamental, context-agnostic functional priors that transfer more robustly to unseen perturbations for zero-shot generalization.
>
> ***Q7: Are the gains statistically significant?***
>
> **A7:** We now report mean ± standard error and paired Wilcoxon tests against the best baseline `CellFlow` [3]; the improvement is supported across perturbations, not just in marginal averages.
>
> >Table 3: Paired comparison vs. `CellFlow`.
>
> |Method|ρΔ|ρΔ^D|ACCΔ|ACCΔ^D|
> |---|---:|---:|---:|---:|
> |CellFlow|0.788 ± 0.002|0.730 ± 0.006|0.913 ± 0.002|0.940 ± 0.005|
> |scBIG|0.846 ± 0.005|0.814 ± 0.013|0.917 ± 0.004|0.990 ± 0.001|
> |Paired Wilcoxon `p`|4.4e-5|0.011|0.043|0.002|
>
> ***Q8: Does `scBIG` generalize beyond genetic perturbations?***
>
> **A8:** **scBIG can indeed generalize to drug perturbations.** We additionally evaluate `scBIG` on the `ComboSciPlex`[4] drug-combination dataset. The model remains strongest overall and improves both correlation and distribution matching over `CPA`[4] and `CellFlow` [3], indicating that the module-level inductive bias transfers beyond CRISPR perturbations.
>
> >Table 4: `ComboSciPlex` drug-combination results.
>
> |Method|ρΔ|ρΔ^D|MSE|Discr. Cos|E-Dist|
> |---|---:|---:|---:|---:|---:|
> |CPA|0.839|0.900|0.0652|0.84|10.2|
> |STATE|0.623|0.742|0.0904|0.62|13.5|
> |CellFlow|0.828|0.919|0.0461|0.82|4.83|
> |scBIG|0.867|0.932|0.0451|0.86|4.74|
>
> This supports cross-context and cross-modality generalization.
>
> [1] Yang et al., *Cell Res.*, 2024.
>
> [2] Lin et al., *Science*, 2023.
>
> [3] Klein et al., bioRxiv, 2025.
>
> [4] Lotfollahi et al., *Mol Syst Biol*, 2023.
>
> ---
> We hope our responses have addressed your concerns and we are more than happy to include all these discussions in the revision.

---

> > ### Author Rebuttal · Reviewer_Vjfe · 2026-04-02
> >
> > I thank the authors for the response. The responses were not exactly for my questions. I maintain my score.

---

> > > ### Author Response · Authors · 2026-04-03
> > >
> > > Thank you for the thoughtful follow-up and for recognizing our work. In our previous response, we mainly used the space for additional experiments, so some clarifications may not have aligned with your questions as directly as they should have. Below we clarify the points that may have remained unclear.
> > >
> > > **1. `I` in Eq. (3) is a model-level learnable bottleneck, and Eq. (4) uses mean pooling because the global interaction has already happened.**
> > >
> > > `I` is initialized once per model, jointly optimized with the encoder, shared across samples, and **not** reinitialized per batch. Its role is not to represent fixed biological modules, but to serve as latent slots in the Perceiver-style bottleneck: `I'` aggregates information from all modules, `I''` models latent interactions, and the updated states are then broadcast back to `H^(1)`. We will add the exact initialization detail in Sec. 2.2.
> > >
> > > After Eq. (3) has modeled the global set interaction, Eq. (4) can use a simple readout. This is also supported by [Table A1](https://anonymous.4open.science/r/scBIG-4A15/README_Vjfe.md): average pooling is stronger than attention pooling and set-transformer pooling (`0.8496/0.8230` vs. `0.8363/0.8124` and `0.8266/0.7971` on `ρΔ/ρΔ^D`). `M` should therefore be viewed as bottleneck capacity rather than a biological quantity; in practice we scale it with `K` (roughly `M ~= K/4`) and choose the **smallest** `K/M` pair near the validation optimum, which is why `M=8` is used in the main setting.
> > >
> > > **2. The balanced constraint should be understood as an architectural bias for stable module reasoning, rather than a biological claim.**
> > >
> > > We agree this distinction should have been clearer. We do not intend the balanced constraint to suggest that true biological programs are equal-sized; rather, balanced clustering helps avoid two failure modes in chunk-wise attention: oversized clusters dilute signal, while tiny clusters lose minority genes. It is an encoder-side regularization choice rather than a biological claim.
> > >
> > > The added comparison helps quantify this trade-off more directly. Under `GRC`, the soft/variable-size version remains competitive, but the hard-balanced version gives slightly better overall metrics and much stronger coherence (`PPI density = 0.224` vs. `0.118`; `Hallmark enrichment = 0.688` vs. `0.281`). For the component gain itself, we do **not** claim a dramatic step change: under the same `GCAE`, moving from `K-means` to `GRC` improves `ρΔ` from `0.8358` to `0.8496`, with partially overlapping `95%` CIs. We therefore interpret the gain as **consistent and complementary**, rather than as a dramatic jump.
> > >
> > > **3. The scaling bottleneck is mainly memory, and the practical compute profile is as follows.**
> > >
> > > On a single RTX 4090 (`24 GB`), the default Norman setting (`2051` HVGs, `K=32`, `M=8`) uses about `4 GB` at batch `256` or `23 GB` at batch `1024`, with step time `87.44 ms`. For `5018` HVGs, the model remains trainable on the same GPU (`8 GB` at batch `256`, or `23 GB` at batch `1024`). For `10000` HVGs, training is still feasible on a `24 GB` GPU by reducing batch size to `128-256`, with `K=32, M=8` using about `17-19 GB` and `K=128, M=32` about `17-22 GB`.
> > >
> > > The practical rule is: keep `K/M` modest for about `2k` HVGs, and for larger gene sets increase `K` and `M` together while reducing batch size. We will move a compact version of these summaries from [Table A3](https://anonymous.4open.science/r/scBIG-4A15/README_Vjfe.md) into the main text and add explicit inference throughput in the revision.
> > >
> > > **4. GeneCompass for clustering and ESM2 for perturbation conditioning reflect a task-specific design choice.**
> > >
> > > The two embeddings are used for **different subproblems**. `GeneCompass` is more suitable for `GRC` because module induction requires context-aware transcriptomic semantics, whereas `ESM2` is more suitable for perturbation conditioning because zero-shot transfer to unseen target genes depends more on intrinsic protein/function priors. Thus, `GeneCompass` is the more helpful prior for discovering coherent modules, whereas `ESM2` is more helpful for encoding perturbation identity under unseen-gene transfer.
> > >
> > > **5. `scBIG` also generalizes beyond the original single-cell-line CRISPR setting.**
> > >
> > > We now additionally report cross-cell-line transfer (train on Replogle_K562, test held-out perturbations in Replogle_RPE1), where the gain is modest but consistent:
> > >
> > > > **Table 5: Cross-cell-line transfer**
> > >
> > > |Method|ρΔ|ρΔ^D|MSE|Disc.Cos|Wass.|
> > > |---|---:|---:|---:|---:|---:|
> > > |CellFlow|0.409|0.531|0.0377|0.756|13.7|
> > > |**scBIG**|**0.420**|**0.555**|**0.0324**|**0.764**|**13.2**|
> > >
> > > Together with the `ComboSciPlex` result (Table 4), this supports both cross-cell-line and cross-modality generalization beyond the original CRISPR setting.
> > >
> > > ---
> > >
> > > We hope these clarifications are helpful and closer to your questions. We sincerely appreciate your careful reading, and if any point is still unclear, we would be happy to clarify it further.

---

### Official Review · Reviewer_Djh4 · 2026-03-13

**Soundness:** 2
**Presentation:** 3
**Significance:** 3
**Originality:** 3
**Overall Recommendation:** 3
**Confidence:** 4

**Summary:**

The paper introduces scBIG, a generative framework for predicting single-cell transcriptional responses to genetic perturbations. The authors argue that existing methods focus on independent genes and fail to capture coordinated, program-level changes. scBIG addresses this by:
- Gene-Relation Clustering (GRC): Inducing biologically coherent gene modules by fusing foundation model embeddings (e.g., GeneCompass) with Protein-Protein Interaction (PPI) priors via Optimal Transport.
- Gene-Cluster-Aware Encoder (GCAE): A hierarchical architecture that models inter-program interactions using a bottleneck attention mechanism.
- Conditional Flow Matching: A generative backbone that models the transition from control to perturbed latent states.
Structure-Aware Alignment: Incorporating cluster correlation and pathway-informed regularizers to ensure biological fidelity in the generated expression.

Experiments on the Norman and RPE1 datasets show scBIG consistently outperforms baselines in both combinatorial and zero-shot (holdout) settings.

**Compliance With Llm Reviewing Policy:**

Affirmed.

**Final Justification:**

After a careful review of the authors and reviewers, I will be maintaining my score. The additional results have added clarity, but overall, I still have hesitations about the impact of the work.

**Key Questions For Authors:**

1. Selection of $z_0$ and $z_1$: How are pairs of control and perturbed cells selected for the training path? Does the model assume any control cell is a valid baseline for any perturbed cell, and if so, how does this account for the inherent heterogeneity within a cell line?
2. Ablation of Priors: Can the authors provide an ablation study where scBIG is trained without foundation model embeddings or PPI priors (e.g., using a random partition or K-means on raw data)? This would clarify if the architecture provides value beyond the priors.
3. Discovery of Novel Coordination: Can the model identify any regulatory coordination that was not already present in the STRING PPI graph or the Reactome database?
4. Static vs. Dynamic Modules: Since clusters are assigned offline and fixed, how does scBIG address its own critique that static priors cannot capture "dynamic program reorganization"?

**Limitations:**

Yes, the authors discussed limitations, specifically regarding the static nature of the gene modules and the limited diversity of the benchmarks used.

**Strengths And Weaknesses:**

Soundness
- Strengths: The use of conditional flow matching is a modern and appropriate choice for modeling continuous state transitions in high-dimensional latent spaces. The extensive benchmarking against many baselines provides a broad performance context.
- Weaknesses: There is a significant circularity in the evaluation. The model uses Reactome pathways and PPI priors to induce modules and regularize the output, and then uses the same biological labels to "validate" the functional relevance of the results. Furthermore, the claim of capturing "dynamic reorganization" is undermined by the fact that gene clusters are fixed offline before training.

Presentation
- Strengths: The framework overview (Figure 2) is detailed and helpful for understanding the multi-stage pipeline. The appendix provides thorough implementation details and hyperparameter settings.
- Weaknesses: The presentation of metrics is poor; several key metrics (e.g., $ACC\Delta^D$, PDS) are used in main-text tables without being defined until the appendix. The Related Work section is disproportionately long, consuming space that could have been used for critical visual results like UMAPs.

Significance
- Strengths: Predicting combinatorial perturbation effects is a high-impact problem in functional genomics. Improving zero-shot generalization is crucial for making in silico models practical for drug discovery.
- Weaknesses: The model appears overengineered. It relies on a dense stack of priors (PPI, Reactome, Foundation Models), making it unclear if the performance gains come from the machine learning architecture or simply the high-quality biological information being "fed" to the model.

Originality
- Strengths: The hierarchical approach of using "inducing points" within a Perceiver-style bottleneck to model interactions between gene clusters is a creative application of attention mechanisms to transcriptomics.
- Weaknesses: While the combination of tools is novel, the individual components (Flow Matching, OT, Foundation Model embeddings) are existing techniques. The "novelty" is primarily a pipeline integration rather than a fundamental algorithmic shift.

Ratings
- Soundness: 2 (Fair). The methodology is mathematically rigorous, but the central claims regarding "dynamic programs" are not supported by the static implementation, and the evaluation lacks biological replicates and non-circular validation.
- Presentation: 3 (Good). The paper is well-structured, but the omission of metric definitions in the main text and the lack of dimensionality reduction plots (UMAPs) hinder immediate understanding.
- Significance: 3 (Good). The problem is vital. Even if the method is an "engineering pipeline," a 6.7% improvement in this domain is non-trivial for practitioners.
- Originality: 3 (Good). The specific way foundation model semantics are fused with PPI priors via OT to create a "module-inductive" bias is a distinct contribution.

---

> ### Author Rebuttal · Authors · 2026-03-27
>
> We thank the reviewer for the constructive feedback and for viewing our inter-gene-cluster modeling as creative. **We believe the clarifications and new results below address concerns and respectfully ask you to reconsider your rating.**
>
> ---
>
> ***Q1: Circular validation?***
>
> **A1:** We respectfully clarify that **`GRC` does not use any Reactome information**. In Sec. 3.4, Reactome is used only to validate the biological relevance of learned `GRC` modules. We will make this clearer in Sec. 3.4.
>
> ***Q2: Fixed modules vs. dynamic programs?***
>
> **A2:** scBIG uses a context-specific but perturbation-stable scaffold: **because HVGs differ across cell lines, `GRC` induces different module partitions for each cell line (See main text Secs. 2.1 and 3.4)**, while perturbation-specific module states remain dynamic.
>
> >Table 1: Dynamic reorganization over a stable scaffold.
>
> |Cell|Perturb.|Top mod|Main genes|
> |---|---|---|---|
> |K562|NCL|M28|CLC, ID1, NFE2, CFD|
> |RPE1|NCL|M7|CDKN2B, CTU2, SESN1|
>
> `Top mod` is the most responsive module under each perturbation. **Even for the same perturbation (`NCL`), K562 and RPE1 activate different top modules**, showing context-specific reorganization over a stable scaffold.
>
> >Table 2: Perturbation-updated clustering during training.
>
> |Update|ρΔ|ρΔ^D|ACCΔ|ACCΔ^D|
> |---|---:|---:|---:|---:|
> |Pert.-updated|0.840|0.807|0.908|0.987|
> |Dataset-HVG(scBIG)|0.850|0.823|0.920|0.991|
>
> We have also explored dynamically reclustering genes during training. Our current results suggest that **`GRC` is more stable than fully updating clusters from perturbation predictions (Table 2)**, and we will continue studying this direction in future work.
>
> ***Q3: Metrics and UMAPs?***
>
> **A3:** Thank you for your suggestions. We add a Norman-additive **UMAP(https://anonymous.4open.science/r/scBIG-4A15/UMAP.png)**, where scBIG outperforms CellFlow[1]. In the revision we will also define metrics concisely in the main text and shorten Related Work.
>
> ***Q4: Are the gains mainly from priors?***
>
> **A4:** **Even without priors, scBIG achieves strong performance**, outperforming the CellFlow baseline on `ρΔ` and `ρΔ^D`. This demonstrates that the architecture itself is fundamentally strong. Nevertheless, **biologically informed priors remain highly meaningful**, as shown in Table 3.
>
> >Table 3: Prior ablation of scBIG
>
> |Method|ρΔ|ρΔ^D|ACCΔ|ACCΔ^D|
> |---|---:|---:|---:|---:|
> |CellFlow|0.789|0.728|0.915|0.939|
> |scBIG w/o any prior|0.824|0.807|0.887|0.966|
> |scBIG w/o GeneCompass|0.833|0.823|0.910|0.991|
> |scBIG w/o PPI|0.840|0.799|0.919|0.981|
> |scBIG w/o pathway loss|0.831|0.815|0.911|0.986|
> |scBIG|0.850|0.823|0.920|0.991|
>
> GeneCompass mainly helps the global metrics, whereas PPI and pathway supervision help the DEG-focused metrics. The full model gives the best overall balance, consistent with complementary rather than redundant priors.
>
> ***Q5: Pipeline integration?***
>
> **A5:** This is not a simple extension. Fusing PPI priors via OT to create the "module-inductive" bias **you acknowledged is a distinct contribution**. Furthermore, as stated in **Sec. 1**, we are the first to systematically frame single-cell perturbation prediction with this **module-level inductive bias**. `GRC` induces programs, `GCAE` models inter-program reasoning, and alignment ensures coherence. We will make this clearer in Sec. 1.
>
> ***Q6: How are `z_0`/`z_1` pairs formed?***
>
> **A6:** **Following standard generative modeling practices like CellFlow** [1], we do not assume one-to-one pairs. Instead, we separately sample control and perturbed cells to construct mini-batch OT pairs, where heterogeneous controls are weighted by proximity in the learned space rather than treated equally. We will make this clearer in Sec. 3.1.
>
> ***Q7: Novel Coordination?***
>
> **A7:** **We indeed identified novel coordination**, which we attribute as a primary source of the performance gains.
>
> >Table 4: Candidate coordination beyond audited priors on Norman. **Module identities are at https://anonymous.4open.science/r/scBIG-4A15/README_Djh4.md.**
>
> |Pair|Perturbation|Coact|Boot|
> |---|---|---:|---:|
> |M2-M30|CEBPE+ZC3HAV1|0.57|1.00|
> |M2-M8|OSR2+UBASH3B|0.52|0.98|
> |M2-M7|OSR2+UBASH3B|0.42|0.97|
>
> `Coact` is positive coactivation across perturbations, and `Boot` its bootstrap positive rate. `CEBPE+ZC3HAV1` aligns with `M2-M30`, while `OSR2+UBASH3B` elevates `M2-M8` and `M2-M30`, indicating recurrent reuse beyond a fixed prior.
>
> For example, `M2` is chromatin/proteostasis-like and `M30` cytoskeleton/growth-like, making `M2-M30` biologically interpretable and consistent with prior chromatin-cytoskeleton coupling work [2]. We therefore view these as **candidate module-level coordination patterns**.
>
> [1] Klein et al., *bioRxiv*, 2025.
>
> [2] Alam et al., *Sci Rep*, 2016.
>
> ---
>
> We are encouraged by your positive view of the paper and **respectfully ask whether these analyses would merit an updated score.** Thank you once again for your time and effort.

---

> > ### Author Rebuttal · Reviewer_Djh4 · 2026-04-02
> >
> > I thank the authors for their responses and for clarifying several of my initial technical concerns. While scBIG represents a sophisticated engineering contribution with impressive performance on standard benchmarks, I remain hesitant regarding its overall biological impact.
> >
> > As noted more explicitly by Reviewer j2pN, the primary challenges currently facing the single-cell perturbation field involve transitioning from immortalized cell lines to complex, multi-cellular systems and heterogeneous donor data. While scBIG achieves strong results on datasets like K562 and RPE1, these environments lack the biological complexity found in more generic frameworks which model whole-organism development or multi-donor primary cells (such as those analyzed in CellFlow). Because the current scope is limited to single cell lines, it remains unclear how these gene programs translate to the more heterogeneous and biologically relevant systems that define the field's current obstacles. I will consider my final score following the responses from the other reviewers.

---

> > > ### Author Response · Authors · 2026-04-02
> > >
> > > We thank the reviewer for the thoughtful follow-up and for acknowledging that initial concerns have been fully addressed. In response to the further comments regarding biological significance and the transition to more heterogeneous systems (a challenge we agree is central to this field), we provide additional point-by-point responses below.
> > >
> > > **1. `scBIG`: A Program-Centric Modeling Principle Prioritizing Coordinated Gene Programs over Independent Genes, Not Merely an Engineering Integration**
> > >
> > > We would like to clarify that the central contribution of `scBIG` is to **introduce and validate a program-centric modeling principle for perturbation response prediction**: cellular responses are better represented as coordinated gene programs than as independent gene-level fluctuations.
> > > By introducing a biologically informed inductive bias in the latent space, `scBIG` moves beyond flat gene-wise prediction and instead captures coordinated regulatory structure, which is especially important for high-dimensional and sparse single-cell data.
> > >
> > > **2. `scBIG` Generalizes Beyond Standard Single-Cell-Line Benchmarks to More Heterogeneous Biological Settings**
> > >
> > > We respectfully note that the K562/RPE1 benchmarks remain important testbeds for evaluating methodological advances under well-characterized settings, as also adopted in recent studies such as `scDFM` [1] (ICLR 2026) and `PRESCRIBE` [2] (NeurIPS 2025). To address the concern about broader biological scope, which was raised by Reviewer `j2pN`, **we had already added two additional evaluations in our response to Reviewer `j2pN`** under more heterogeneous transfer settings, and summarize them here for completeness:
> > >
> > > **i) Cross-cell-line transfer.** We trained on Replogle_K562 and tested on held-out perturbations in Replogle_RPE1.
> > >
> > > > **Table 1: Cross-cell-line transfer**
> > >
> > > |Method|ρΔ|ρΔ^D|MSE|Disc.Cos|E-Dist|Wass.|
> > > |---|---:|---:|---:|---:|---:|---:|
> > > |CellFlow|0.409|0.531|0.0377|0.756|6.51|13.7|
> > > |**scBIG**|**0.420**|**0.555**|**0.0324**|**0.764**|**5.37**|**13.2**|
> > >
> > > Again, `scBIG` remains stronger than `CellFlow`, suggesting that its program-level inductive bias remains beneficial when the cellular background changes.
> > >
> > > **ii) Unseen-drug generalization.** We also tested unseen-drug prediction on ComboSciPlex [3], using the same ChemBERTa [4] drug encoder for all methods.
> > >
> > > > **Table 2: Unseen-drug prediction on ComboSciPlex**
> > >
> > > |Method|ρΔ|ρΔ^D|MSE|Disc.Cos|E-Dist|Wass.|
> > > |---|---:|---:|---:|---:|---:|---:|
> > > |scGPT|0.817|0.928|0.0597|0.803|14.1|17.5|
> > > |CPA|0.839|0.900|0.0652|0.839|10.2|18.2|
> > > |STATE|0.623|0.742|0.0904|0.615|13.5|17.9|
> > > |CellFlow|0.828|0.919|0.0461|0.818|4.83|17.5|
> > > |**scBIG**|**0.867**|**0.932**|**0.0451**|**0.861**|**4.74**|**16.8**|
> > >
> > > `scBIG` achieves the best overall performance and outperforms both `CellFlow` and `CPA` [3] on mean-level and distribution-level metrics.
> > >
> > > Regarding the cross-donor setting in `CellFlow`, we began processing the PBMC dataset [5] and training the corresponding models **immediately after seeing your comment**. Because the discussion period is **limited**, this experiment is not yet complete. Nevertheless, given the consistent gains of `scBIG` in other heterogeneous transfer settings, including multi-cell-line unseen-perturbation prediction, cross-cell-line transfer, and unseen-drug prediction, we are confident it will outperform the `CellFlow` baseline here as well, and we will include this experiment in the revised version.
> > >
> > > Taken together, these results suggest that **the benefit of `scBIG` is not confined to a single benchmark or immortalized cell line**. They further indicate that program-level modeling yields a more transferable representation under cell-line and perturbation shifts, supporting `scBIG` as a biologically meaningful and transferable program-centric framework.
> > >
> > > **3. Broader significance**
> > >
> > > While we do not claim that this paper **fully solves** perturbation modeling in the most complex multicellular settings, the new results support a broader possibility: coordinated gene programs may be a **more stable and transferable unit** than isolated genes. We view this as the main biological insight of `scBIG` and will revise the manuscript to state this scope more precisely.
> > >
> > > [1] Yu et al., *arXiv*, 2026.
> > >
> > > [2] Cheng et al., *arXiv*, 2025.
> > >
> > > [3] Lotfollahi et al., *Mol Syst Biol*, 2023.
> > >
> > > [4] Chithrananda et al., *arXiv*, 2020.
> > >
> > > [5] Parse Biosciences, *10 Million Human PBMCs in a Single Experiment*, https://www.parsebiosciences.com/datasets/10-million-human-pbmcs-in-a-single-experiment, 2025.
> > >
> > > ---
> > >
> > > Overall, we hope these clarifications and additional results address the concern. While more complex multicellular and multi-donor settings remain important future directions, we believe this work makes a meaningful contribution by **establishing a biologically grounded program-centric modeling principle within its stated scope**. We thank the reviewer again and will revise the manuscript to make this scope and contribution clearer.

---

### Official Review · Reviewer_UrBu · 2026-03-13

**Soundness:** 4
**Presentation:** 4
**Significance:** 3
**Originality:** 3
**Overall Recommendation:** 4
**Confidence:** 3

**Summary:**

* $\textbf{Problem}$ : Existing gene perturbation prediction methods model genes independently, which fails to capture the coordinated responses that occur at the level of functional gene modules in real biological systems.
* $\textbf{Method}$ : To address this, the authors group genes into functional modules, learn module-level representations with a module-aware encoder, and model the transition from control to perturbation states using conditional flow matching with additional biological regularization.
* $\textbf{Conclusion}$ : Modeling gene expression with module-level structural representations improves perturbation prediction performance, particularly for unseen and combinatorial perturbations.

**Compliance With Llm Reviewing Policy:**

Affirmed.

**Final Justification:**

The authors made substantial efforts to address my concerns, and most of them have been resolved. Therefore, I maintain my score, which supports acceptance.

**Key Questions For Authors:**

* How sensitive is the model’s performance to the number of gene modules and the specific clustering strategy, given that the modules are constructed using pretrained gene embeddings and PPI networks?
* The proposed method consists of several components such as module clustering, a Perceiver-based encoder, flow matching, and biological regularization, making it difficult to clearly determine how much each component contributes to the overall performance improvement.
* Because gene modules are constructed through a predefined clustering step and used as a fixed structure, the model may not fully capture gene interactions or gene programs that vary depending on the biological context.

**Limitations:**

yes

**Strengths And Weaknesses:**

###  Soundness
* strength :  The proposed method is generally technically sound and supported by experiments on multiple perturbation datasets

### Presentation
* strength : The paper is mostly well organized and clearly motivates the shift from gene-level to module-level modeling

### Significance
* strength : Gene perturbation prediction is an important task for single-cell biology and drug discovery, making improvements in this area valuable.

### Originality
* strength : The paper offers a novel combination of module-based representations and generative perturbation modeling that provides a new perspective on the problem.

---

> ### Author Rebuttal · Authors · 2026-03-27
>
> We sincerely thank Reviewer UrBu for the positive and insightful feedback. We are encouraged that you recognize our work as offering a new perspective through gene modules for perturbation prediction. Below we address your concerns point by point.
>
> ---
> ***Q1: K/clustering sensitivity?***
>
> **A1:** Thank you for your suggestions. **We included ablations on `K` and clustering in Fig. 6 and Table 4 (main text)**. Here we additionally report results for `K=64/128` and finer GRC ablations. `K=32` is a strong default for about **2k HVGs**; setting `K` too large causes a modest performance drop. **[Table A1(Link)](https://anonymous.4open.science/r/scBIG-4A15/README_UrBu.md)** further shows that `K=64` is a good upgrade for about **5k HVGs**, while `K=128` works slightly better for about **10k HVGs**.
>
> >Table 1: `K` sensitivity.
>
> |K|ρΔ|ρΔ^D|ACCΔ|ACCΔ^D|
> |---|---:|---:|---:|---:|
> |16|0.837|0.830|0.903|0.980|
> |32|0.850|0.823|0.920|0.991|
> |64|0.814|0.805|0.912|0.975|
> |128|0.794|0.793|0.899|0.970|
>
> >Table 2: Clustering sensitivity.
>
> |Method|ρΔ|ρΔ^D|ACCΔ|ACCΔ^D|
> |---|---:|---:|---:|---:|
> |No clustering|0.808|0.811|0.901|0.963|
> |Random partition|0.820|0.773|0.902|0.967|
> |K-means (hard)|0.836|0.819|0.916|0.981|
> |K-means (soft)|0.825|0.777|0.916|0.972|
> |GRC w/o GC|0.833|0.823|0.910|0.991|
> |GRC w/o PPI|0.840|0.799|0.919|0.981|
> |GRC (soft)|0.832|0.828|0.915|0.984|
> |GRC|0.850|0.823|0.920|0.991|
>
> These results show that **gene grouping helps** (`K-means` > `Random partition`), but **biologically informed `GRC` performs best**. `PPI` mainly improves DEG metrics, `GC` mainly improves global metrics, and hard `GRC` slightly outperforms soft `GRC`, suggesting that **balanced assignments** stabilize cross-module attention. **We will make this clearer in Sec. 3.5.**
>
> ***Q2: Contribution of each component?***
>
> **A2:** **Table 4(a, c) in the paper already presents ablations. For clarity, we summarize the gain from each component below.**
>
> >Table 3: Component ablation.
>
> |Model|ρΔ|ρΔ^D|ρΔ drop (%)|
> |---|---:|---:|---:|
> |w/o GRC|0.808|0.811|4.92|
> |w/o GCAE|0.824|0.805|3.02|
> |w/o pathway loss|0.831|0.815|2.24|
> |w/o corr loss|0.836|0.805|1.59|
> |w/o two losses|0.824|0.801|2.99|
> |scBIG|0.850|0.823|-|
>
> **`GRC` yields the largest gain (4.92% in `ρD`), highlighting the value of module-level modeling.** The Perceiver-based encoder (`GCAE`) improves `ρD` by `3.02%`, while biological regularization contributes `2.99%`. **We will add this analysis to the revised version.**
>
> ***Q3: Can fixed modules capture context-dependent programs?***
>
> **A3:** Yes，scBIG can indeed capture context-dependent programs: **because HVGs differ across cell lines, `GRC` induces different module partitions for each cell line (See main text Sec. 2.1 and 3.4)**, while perturbation-specific module states remain dynamic.
>
> >Table 4: Dynamic reorganization over a stable scaffold.
>
> |Cell|Perturb.|Top mod|Main genes|
> |---|---|---|---|
> |K562|NCL|M28|CLC, ID1, NFE2, CFD|
> |RPE1|NCL|M7|CDKN2B, CTU2, SESN1|
>
> `Top mod` is the most responsive module under each perturbation. **Even for the same perturbation (`NCL`), K562 and RPE1 activate different top modules**, showing context-specific reorganization over a stable scaffold.
>
> >Table 5: Perturbation-updated clustering during training.
>
> |Update|ρΔ|ρΔ^D|ACCΔ|ACCΔ^D|
> |---|---:|---:|---:|---:|
> |Pert.-updated|0.840|0.807|0.908|0.987|
> |Dataset-HVG(scBIG)|0.850|0.823|0.920|0.991|
>
> We have also explored dynamically reclustering genes during training. Our current results suggest that **`GRC` is more stable than fully updating clusters from perturbation predictions (Table2), and we will continue studying this direction in future work.**
>
> In addition, we observed an interesting phenomenon: **the model identifies candidate cross-module coordination patterns under specific perturbations, beyond the fixed prior.**
>
> >Table 6: Candidate cross-module coordination. Module identities are listed in **[Table A2(Link)](https://anonymous.4open.science/r/scBIG-4A15/README_UrBu.md)**
>
> |Pair|Perturb.|Coact|Boot|
> |---|---|---:|---:|
> |M2-M30|CEBPE+ZC3HAV1|0.568|1.000|
> |M2-M8|OSR2+UBASH3B|0.515|0.980|
> |M2-M7|OSR2+UBASH3B|0.423|0.970|
>
> `Coact` is positive coactivation across perturbations and `Boot` its bootstrap positive rate.
>
> **For `CEBPE+ZC3HAV1`, `M2` is chromatin/proteostasis-like and `M30` cytoskeleton/growth-like, suggesting coupled differentiation/stress rewiring and a downstream cytoskeletal-growth response, consistent with prior chromatin-cytoskeleton coupling work[1].**
>
> **This provides stronger validation that GRC captures dynamic module coordination, and we will emphasize this point in Sec. 3.4.**
>
> [1] Alam et al., *Sci Rep*, 2016.
>
> ---
> We hope these clarifications address your concerns and will be incorporated into the revision. If you have any additional concerns or comments that we may have missed in our responses, we would be most grateful for any further feedback from you to help us further enhance our work.

---

> > ### Author Rebuttal · Reviewer_UrBu · 2026-04-01
> >
> > Most of my concerns have been addressed, and therefore, I will maintain my score

---

> > > ### Author Response · Authors · 2026-04-01
> > >
> > > Dear Reviewer UrBu,
> > >
> > > We hope this message finds you well. We sincerely thank you for your positive and constructive feedback on our manuscript. Your overall assessment is profoundly meaningful to our research team, serving as **a lantern in the dark to inspire us to continuously refine and advance this line of research.**
> > >
> > > Your insightful comment regarding whether **scBIG can capture gene modules across diverse biological contexts** is exceptionally valuable to our work. We will supplement detailed targeted analyses for this comment in Section 3.4 of our revised manuscript. Specifically, we will present the module response patterns of both identical and distinct perturbations across independent datasets, as well as the novel module coordination behaviors captured by scBIG.
> > >
> > > Should you have any additional comments, suggestions or feedback for us to address, please do not hesitate to let us know. **Your insights are invaluable to us**, and we are fully committed to resolving all remaining concerns to further improve the rigor and quality of our study.
> > >
> > > We greatly appreciate the time and effort you have dedicated to reviewing our manuscript.

---

### Decision · Program_Chairs · 2026-04-30

**Decision:**

Accept (regular)

**Comment:**

The paper proposes scBIG, a module-inductive generative framework for predicting transcriptional responses to genetic perturbations. The framework's key concept is to move beyond gene-wise modelling by explicitly inducing coordinated gene programs

The reviewers are mostly positive about this work. They all agree that the paper addresses a challenging problem in single-cell biology; that the method is novel and well designed; and that the core idea is well grounded in biology. The experiments in the paper are extensive, and scBIG achieves state-of-the-art performance against a large number of baseline methods. The impact of its core components is studied through a comprehensive set of ablation experiments that were further extended during the rebuttal process.

The reviewers identified several issues, such as presentation problems, circular evaluation and missing experiments, that were all convincingly addressed during the rebuttal. Reviewer Djh4 still has one remaining concern about the overall biological impact of the work and whether it would transfer to more heterogeneous biological settings. In response, the authors conducted some preliminary experiments in more heterogeneous settings, but argued that this would be beyond the scope of their claimed contribution, which seems a reasonable response to me.

Overall, I believe that the paper makes a novel, solid and well-evaluated contribution to the machine learning in genomics community, and I therefore recommend its acceptance. I urge the authors to implement all the changes they promised in their rebuttal.